# TRANSFERABLE UNLEARNABLE EXAMPLES

**Jie Ren**[*]
Michigan State University
`renjie3@msu.edu`

**Han Xu**[*]
Michigan State University
`xuhan1@msu.edu`

**Yuxuan Wan**
Michigan State University
`wanyuxua@msu.edu`

**Xingjun Ma**
Fudan University
`xingjunma@fudan.edu.cn`

**Lichao Sun**
Lehigh University
`lis221@lehigh.edu`

**Jiliang Tang**
Michigan State University
`tangjili@msu.edu`

## ABSTRACT

With more people publishing their personal data online, unauthorized data usage has become a serious concern. The *unlearnable examples* strategies have been introduced to prevent third parties from training on the data without permission. They add perturbations to the users' data before publishing, so as to make the models trained on the perturbed published dataset invalidated. These perturbations have been generated for a specific training setting and a target dataset. However, their unlearnable effects significantly decrease when used in other training settings or datasets. To tackle this issue, we propose a novel unlearnable strategy based on *Class-wise Separability Discriminant* (CSD), which boosts the transferability of the unlearnable perturbations by enhancing the linear separability. Extensive experiments demonstrate the transferability of the unlearnable examples crafted by our proposed method across training settings and datasets. The implementation of our method is available at https://github.com/renjie3/TUE.

## 1 INTRODUCTION

With more people posting their personal data online (intentionally or unintentionally), it has raised the concern that the data might be utilized without the owner's consent to train commercial or malicious machine learning models. While large-scale datasets collected from the Internet like LFW(Huang et al., 2008), Freebase(Bollacker et al., 2008), and Ms-celeb-1m(Guo et al., 2016) have greatly advanced the development of deep learning, they may contain a certain amount of private data, which has the potential risk of privacy leakage. Thus, growing efforts (Huang et al., 2020; Fowl et al., 2021) have been made to protect data from unauthorized usage by making the data samples "unlearnable" (Huang et al., 2020; Fowl et al., 2021; He et al., 2022). These methods generate the unlearnable examples by injecting imperceptible "shortcut" perturbation. If the data is used by unauthorized training, the models will be tricked to extract such easy-to-learn shortcut features and ignore the real semantics in the original data (Geirhos et al., 2020). Consequently, the trained model fails to recognize the user's data during the test phase and the user's data gets protected.

As shown in Figure 1, the practical usage procedure of unlearnable examples (UEs) consists of two stages, i.e., the generation stage and the evaluation stage. Before releasing data, in the *generation* stage, one can make the data unlearnable by adding perturbations into original data. According to whether the label information is used to generate the perturbations, we divide the UEs into **Supervised UEs** which is generated with the guidance of the label information, and **Unsupervised UEs** which is generated without the guidance of label information. In the *evaluation* stage, the unlearnable version of data is released to the public, and the unauthorized third parties might use different algorithms, like supervised training and unsupervised training, to learn from the data. The UEs aim to provide protection in this stage against unauthorized training and invalidate the models trained on them.

However, existing UEs are in lack of training-wise transferability and have weaknesses in data-wise transferability in the evaluation stage. **First,** low training-wise transferability implies that the perturbed samples generated towards one target training setting cannot be transferred to other

---

[*]Equal contribution.

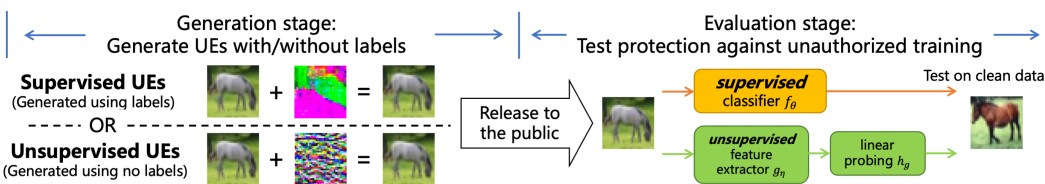

Figure 1: Two stages of Unlearnable Examples.

training settings. As shown in Section 3.2.1, although Error-Minimizing Noise (EMN) (Huang et al., 2020) can protect data from supervised training, we can use unsupervised learning methods, such as Contrastive Learning (Chen et al., 2020a; Chen & He, 2021; Chen et al., 2020b), to extract useful representations from the EMN-protected dataset and obtain high accuracy. In this way, the perturbed dataset is still exposed to no protection. **Second**, insufficient data-wise transferability indicates that the unlearnable effect of perturbations generated for one target dataset will significantly decrease when transferred to other datasets. We have to generate perturbations for each dataset, reducing its flexibility in real-world scenarios. For example, data in various applications such as social media is dynamic or even streaming, which makes it extremely challenging to generate the entire perturbation set when new data is continuously emerging.

In this work, we aim to enhance the training-wise and data-wise transferability of unlearnable examples. In detail, our method is motivated by Synthetic Noise (SN) (Yu et al., 2021), which devises a manually designed linear separable perturbation to generate UEs. Such perturbations do not target specific dataset, thus it has the potential to enhance data-wise transferability. However, SN is manually designed and it is not quantifiable or optimizable. As such, it is impossible to incorporate SN into other optimization processes. Meanwhile, SN lacks training-wise transferability. Therefore, in this work, we propose *Class-wise Separability Discriminant* (CSD) to generate optimizable linear-separable perturbations. Our framework *Transferable Unlearnable Examples* with enhanced linear separability can generate UEs with superior training-wise and data-wise transferability.

## 2 RELATED WORK

**Unlearnable Examples.** Unlearnable examples are close to availability attack which aims at making the data out of service for training the models by the unauthorized third parties (Muñoz-González et al., 2017). Several supervised UEs are proposed (Huang et al., 2020; Fowl et al., 2021; Shan et al., 2020). The vanilla unlearnable examples are crafted by generating and adding unlearnable perturbations into the clean samples (Huang et al., 2020). Being induced to trust that the perturbation can minimize the loss better than the original image features, the model will pay more attention to the perturbations. Unlearnable Contrastive Learning (UCL) is proposed (He et al., 2022) to generate UEs to protect data from unsupervised learning. Our studies show that both Supervised and Unsupervised UEs lack training-wise transferability as shown in Section 3.2.1 and Appendix A, respectively. Table 1 summarizes the settings of existing methods, where ✗ means ineffective protection.

Table 1: Unlearnable Examples Settings of Existing Approaches

| Generation \ Evaluation | Supervised Training | Unsupervised Training |
|---|---|---|
| Supervised Unlearnability | (Huang et al., 2020; Fowl et al., 2021) | ✗ |
| Unsupervised Unlearnability | ✗ | (He et al., 2022) |

**Unsupervised Learning.** Recently, unsupervised learning has shown its great potential to learn the representation from unlabeled data. Contrastive learning, one of the popular unsupervised methods, uses the task of instance discrimination to learn the representations. In SimCLR (He et al., 2022) which is the most common contrastive learning method, the positive and negative samples for each instance are created and the task is to discriminate the positive samples and negative samples. Some methods like SimSiam (Chen & He, 2021) and BYOL (Grill et al., 2020) can remove the negative samples and only focus on pushing the representations between positive samples to be similar.

## 3 PRELIMINARY

In this section, preliminary studies are conducted to explore two types of transferability. We first introduce key notations and definitions, and then show the insufficiency of transferability. **Since the majority of UEs are guided by label information, in this work, we focus on Supervised UEs**

**and leave Unsupervised UEs as one future work.** Note that the observations of Supervised UEs in terms of transferability could be applicable to Unsupervised UEs. For example, in Appendix A, we show that Unsupervised UEs also lack training-wise transferability.

### 3.1 DEFINITIONS

In this subsection, we give the definition of UEs and then define the two types of transferability.

**Unlearnable Examples.** Suppose that the dataset contains $n$ clean examples $\mathcal{D}_c = \{(\boldsymbol{x}_i, y_i)\}_{i=1}^n$ with the input data $\boldsymbol{x}_i \in \mathcal{X} \subset \mathbb{R}^d$ and the associated label $y_i \in \mathcal{Y} = \{1, 2, \ldots, K\}$. We assume that the unauthorized parties will use the published data to train a classifier $f_\theta : \mathcal{X} \to \mathcal{Y}$, where $\theta$ is the model parameters, and do inference on their test dataset. To protect the data from unauthorized training, instead of publishing $\mathcal{D}_c$, we generate an unlearnable dataset as $\mathcal{D}_u = \{(\boldsymbol{x}_i + \boldsymbol{\delta}_i, y_i)\}_{i=1}^n$ where $\boldsymbol{\delta}_i \in \Delta_{\mathcal{D}_c} \subset \mathbb{R}^d$ and $\Delta_{\mathcal{D}_c}$ is the perturbation set for $\mathcal{D}_c$. The goal of unlearnability is that if we only publish $\mathcal{D}_u$, the model $f_\theta$ trained on $\mathcal{D}_u$ performs poorly on the test dataset. Benefiting from the constraint $\|\boldsymbol{\delta}\|_p \leq \epsilon$, $\mathcal{D}_u$ appears the same as $\mathcal{D}_c$ in human eyes. The perturbed example $\boldsymbol{x}_i + \boldsymbol{\delta}_i$ is known as the *unlearnable example*. The most representative EMN (Huang et al., 2020) generates supervised UEs by alternating optimization on the bi-level problem:

$$\min_{\theta} \min_{\boldsymbol{\delta}_i \in \{\boldsymbol{v} : \|\boldsymbol{v}\|_\infty \leq \epsilon\}} \sum_{i=1}^n \mathcal{L}\left(f_\theta\left(\boldsymbol{x}_i + \boldsymbol{\delta}_i\right), y_i\right). \tag{1}$$

The outer minimization can imitate the training process, while the inner minimization can induce $\boldsymbol{\delta}_i$ to have the property of minimizing the supervised loss. Due to this property, deep models will pay more attention to the easy-to-learn $\boldsymbol{\delta}_i$ and ignore $\boldsymbol{x}_i$.

**Training-wise Transferability.** Supervised unlearnable examples have been designed to protect data from supervised training in the evaluation stage (Huang et al., 2020). However, the unlearnable effect is almost lost if they are utilized for unsupervised training. The unauthorized parties can first get a useful feature extractor $g_\eta$ from $\mathcal{D}_u$ by unsupervised training like Contrastive Learning (Chen et al., 2020a) and then fine-tunes on $\mathcal{D}_u$ or other data to get the classifier $h_g$. The training-wise transferability means that supervised unlearnable examples can invalidate $g_\eta$ when transferred into unsupervised training.

**Data-wise Transferability.** When a second dataset, $\mathcal{D}_{\tilde{c}} = \{(\tilde{\boldsymbol{x}}_i, \tilde{y}_i)\}_{i=1}^{\tilde{n}}$, where $\tilde{\boldsymbol{x}}_i \in \tilde{\mathcal{X}} \subset \mathbb{R}^d$ and $\tilde{y}_i \in \tilde{\mathcal{Y}} = \{1, 2, \ldots, \tilde{K}\}$, also requires protection, it is more efficient and practical if we can transfer the perturbation that has already been generated for $\mathcal{D}_c$ onto $\mathcal{D}_{\tilde{c}}$. If the perturbation, $\Delta_{\mathcal{D}_c}$, is data-wise transferable, we can quickly create an unlearnable dataset $\mathcal{D}_{\tilde{u}}$ to replace $\mathcal{D}_{\tilde{c}}$ before publishing as $\mathcal{D}_{\tilde{u}} = \{(\tilde{\boldsymbol{x}}_i + \boldsymbol{\delta}_{H(i)}, \tilde{y}_i)\}_{i=1}^n$, where $\tilde{\boldsymbol{x}}_i \in \mathcal{D}_{\tilde{c}}$, $\boldsymbol{\delta}_{H(i)} \in \Delta_{\mathcal{D}_c}$, and $H(i)$ decides which perturbation in $\Delta_{\mathcal{D}_c}$ is added on $\tilde{\boldsymbol{x}}_i$ in the new dataset, $\mathcal{D}_{\tilde{c}}$. Without retraining the perturbation set, the unseen $\mathcal{D}_{\tilde{c}}$ is also protected by transferring the unlearnable perturbations from $\Delta_{\mathcal{D}_c}$.

### 3.2 TRANSFERABILITY IN EXISTING METHODS

In this subsection, we show that existing supervised unlearnable examples have almost no training-wise transferability and insufficient data-wise transferability.

#### 3.2.1 TRAINING-WISE TRANSFERABILITY

In Fig. 2, the experiments demonstrate the ineffective protection of two existing Supervised UEs, i.e. Error-Minimizing Noise (EMN) (Huang et al., 2020) and Synthetic Noise (SN) (Yu et al., 2021) when they are transferred into unsupervised training in the evaluaton stage. The UEs for CIFAR-10 (Krizhevsky et al., 2009) are generated and then applied to prevent either supervised (standard training with cross-entropy loss) or unsupervised (SimCLR plus linear

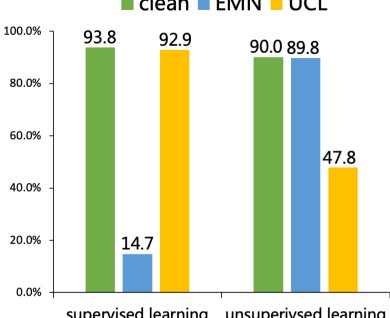

Figure 2: Acc. on clean test data of supervised and unsupervised models trained on clean and EMN-pertubed CIFAR-10

probing (Chen et al., 2020a)) training of a ResNet-18 (He et al., 2016). Fig. 2 shows the test accuracy (accuracy on the original clean test set) of the target models obtained under different training settings. An extremely low test accuracy indicates successful unlearnable protection. EMN decreases the accuracy of supervised training from 93.8% to 14.7%, but only decreases the accuracy of unsupervised training by 0.2%. This suggests that EMN has almost no unlearnable effect under unsupervised learning and thus has extremely low training-wise transferability, which is also the case for SN.

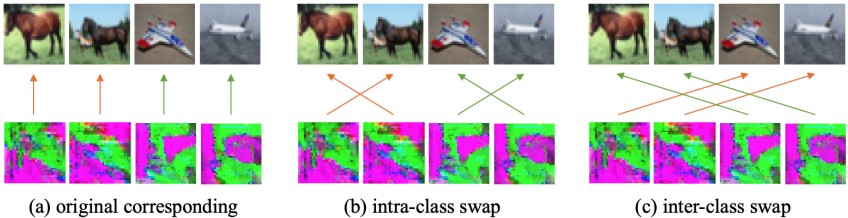

(a) original corresponding    (b) intra-class swap    (c) inter-class swap

Figure 3: Original one-to-one correspondence and two swapping methods.

### 3.2.2 DATA-WISE TRANSFERABILITY

In this subsection, we investigate data-wise transferability for EMN and SN. As introduced in Eq. 1, EMN has been designed to induce $\delta_i$ to minimize the loss in supervised training on $x_i$. Therefore, EMN generates UEs based on target data. Next, we will show the data-wise transferability of EMN on non-target data samples and non-target datasets. For $\delta_i$, the target sample is $x_i$, while the non-target samples are all the other samples $x_j$ in $\mathcal{D}_c$, where $j \neq i$. For the whole perturbation set $\Delta_{\mathcal{D}_c}$, the non-target dataset is any other dataset $\mathcal{D}_{\tilde{c}}$. Since $\delta_i$ focuses on the target sample $x_i$, we will first show the reduction of unlearnable effect on non-target samples. Then we will demonstrate that the perturbation set $\Delta_{\mathcal{D}_c}$ for $\mathcal{D}_c$ cannot hold the strong unlearnable effect on non-target dataset, $\mathcal{D}_{\tilde{c}}$.

On non-target samples, we change the assignment between perturbations and samples by intra-class swapping and inter-class swapping as shown in Fig. 3. Specifically, for inter-class swapping, we apply the unlearnable noise generated for sample $x_i$ to a different sample $x_j$ from a different class (i.e., $y_i \neq y_j$) via random permutation. For intra-class swapping, we swap the noise between samples from the same class also via random permutation. The two sample transferability swaps are illustrated in Fig. 3. As shown in Table 2, the unlearnable effect decreases significantly under both intra-class and inter-class swapping.

On non-target datasets, EMN is also limited in data-wise transferability. We first generate the perturbations $\Delta_{\mathcal{D}_c}$ on CIFAR-10 and then choose another non-target dataset SVHN-small, which is downsampled from SVHN(Netzer et al., 2011), as non-target dataset $\mathcal{D}_{\tilde{c}}$ to protect. CIFAR-10 has ten classes, where every class has 5,000 training images. SVHN also has ten classes, but some classes have more than 5,000 training images. Thus we sample 5,000 training images in every class from SVHN and construct SVHN-small. In Table 3, we first generate EMN on SVHN-small to protect target dataset, i.e. SVHN-small. It protects the data from unauthorized training and reduces the test accuracy to only 11.64%. But when we use EMN generated on CIFAR-10 to protect SVHN-small, SVHN-small becomes a non-target dataset. More information is learned from SVHN-small and the test accuracy is 27.59%, which means the non-target dataset gets less protection.

Table 2: Comparison of supervised unlearnable examples on target and non-target samples.

| Corresponding | CIFAR10 | | CIFAR100 | |
|---|---|---|---|---|
| | EMN | SN | EMN | SN |
| Original | 15.88 | 14.07 | 6.59 | 2.13 |
| Intra-class swap | 30.74 | 13.59 | 21.63 | 2.44 |
| Inter-class swap | 30.15 | 13.15 | 35.50 | 2.73 |

Table 3: Comparison of supervised unlearnable examples on target and non-target datasets.

| Generated on | Tested on SVHN-small | |
|---|---|---|
| | EMN | SN |
| SVHN-small | 11.64 | 8.46 |
| CIFAR-10 | 27.59 | 9.58 |

We conducted similar experiments for SN (Yu et al., 2021). The results for non-target samples and non-target datasets are illustrated in Table 2 and Table 3, respectively. We observe that SN is data-wise transferable. According to (Yu et al., 2021), SN can create linear separability that the label-related imperceptible perturbation is linearly separable between different classes, as a shortcut for the optimization objective. In other words, the linear separability is only between $\delta_i$ and $y_i$, and not related to $x_i$. This indicates the importance of linear separability for data-wise transferability. However, SN is generated by sampling from a manually designed distribution and the sampling process is incompatible with an optimization process. Thus, it is hard to directly incorporate SN into other optimization objectives to enjoy linear separability for data-wise transferability.

### 3.3 DISCUSSIONS

As shown by the above preliminary studies, both EMN and SN lack training-wise transferability, and EMN has insufficient data-wise transferability. Meanwhile, we found that the linearly separable SN perturbation that is independent on the data has good data-wise transferability. However, the manually designed linear separability of SN is unable to be leveraged by other optimization objectives. Thus in

the next section, we propose Class-wise Separability Discriminant and a new framework to generate unlearnable examples with training-wise and data-wise transferability.

# 4 TRANSFERABLE UNLEARNABILITY FROM CLASSWISE SEPARABILITY DISCRIMINANT

In order to improve the two types of transferability of unlearnable examples, we propose the optimizable Class-wise Separability Discriminant (CSD) to quantify linear separability. Furthermore, we propose Transferable Unlearnable Examples (TUE) that have superior training-wise and data-wise transferability. In the evalaution stage, it not only generalizes the protection scenario from supervised to unsupervised training but also maintains the unlearnability when transferred to non-target datasets.

## 4.1 CLASSWISE SEPARABILITY DISCRIMINANT

Linear separability lies in two factors, i.e., intra-class distance and inter-class distance. Intuitively, better linear separability means smaller intra-class distance and larger inter-class distance. In the context of UEs, smaller intra-class distance means that perturbations from the same class concentrate on a small area in the input space, while larger inter-class distance means that perturbations of different classes are far away from each other. When the overlapping area between different classes is reduced by smaller intra-class distance and larger inter-class distance, the perturbations can become features that are easily separated by even a linear classifier. For measuring the intra-class and inter-class distances, we first define the centroid of the perturbations for one class: $c_k = \frac{1}{|\{\delta_i : y_i = k\}|} \sum_{\{\delta_i : y_i = k\}} \delta_i$, where $\{\delta_i : y_i = k\}$ are the perturbations for class $k$. Then we have the intra-class distance as:

$$\sigma_k = \frac{1}{|\{\delta_i : y_i = k\}|} \sum_{\{\delta_i : y_i = k\}} d(\delta_i, c_k), \tag{2}$$

where $d(\cdot, \cdot)$ is the Euclidean distance. $\sigma_k$ measures the average distance between the perturbation $\delta_i$ whose label is $k$ to the centroid $c_k$. The inter-class distance is defined by the Euclidean distance between two centroids:

$$d_{i,j} = d(c_i, c_j) \tag{3}$$

To enhance the linear separability with an optimizable objective function, we propose Class-wise Separability Discriminant (CSD):

$$\mathcal{L}_{\text{CSD}}(\{\delta_i, y_i\}_{i=1}^n) = \frac{1}{M} \sum_{i=1}^{M} \frac{1}{M-1} \sum_{j \neq i}^{M-1} \left( \frac{\sigma_i + \sigma_j}{d_{i,j}} \right), \tag{4}$$

where $M$ is the number of classes, and $y_i$ represents the ground truth label for sample $x_i \in \mathcal{D}_c$. It is worth mentioning that $\mathcal{L}_{\text{CSD}}(\{\delta_i, y_i\}_{i=1}^n)$ is not related to $x_i$, which means that $\delta_i$ is generated independently on $x_i$. $\sigma_i$ and $\sigma_j$ measure the intra-class distance of class $i$ and class $j$, respectively; $d_{i,j}$ measures the inter-class distance between class $i$ and class $j$. The ratio between intra-class distance and inter-class distance, $\frac{\sigma_i + \sigma_j}{d_{i,j}}$, defines the overlapping between two classes. Better clustering distribution has smaller intra-class distance and larger inter-class distance, leading to a lower ratio and smaller overlapping area between two classes. Therefore, a smaller CSD value indicates more compact classes and separable clusters, i.e., better linear separability.

## 4.2 TRANSFERABLE UNLEARNABLE EXAMPLES

As discussed in Section 3.2.2, linear separability plays an essential role in supervised unlearnability and data-wise transferability. In order to enhance the data-wise and training-wise transferability of UEs simultaneously, we incorporate linear separability into unsupervised unlearnable examples, resultign in the Transferable Unlearnable Examples (TUE) framework. In particular, TUE uses contrastive learning as the unsupervised backbone and embeds linear separability via CSD into unsupervised unlearnability. Since TUE uses label information in CSD, it is one kind of supervised unlearnable example methods. Most contrastive learning algorithms, such as SimCLR (Chen et al., 2020a), MoCo (Chen et al., 2020b) and SimSiam (Chen & He, 2021), enforce the similarity between two augmentations of the same samples to learn a representation. TUE generates the perturbation that not only promotes this similarity but also has linear separability, via the following bi-level optimization problem:

$$\min_{\theta} \min_{\{\boldsymbol{\delta}_i : \|\boldsymbol{\delta}_i\|_{\infty} \leq \epsilon\}} \sum_{i=1}^{n} \mathcal{L}_{\mathrm{CL}}\Big(f\big(\theta, T_1(\boldsymbol{x}_i + \boldsymbol{\delta}_i)\big), f\big(\theta, T_2(\boldsymbol{x}_i + \boldsymbol{\delta}_i)\big)\Big) + \lambda \mathcal{L}_{\mathrm{CSD}}(\{\boldsymbol{\delta}_i, y_i\}_{i=1}^{n}), \quad (5)$$

where $\mathcal{L}_{\mathrm{CL}}$ is the loss of contrastive learning, and $\lambda$ is the weight to balance between two loss terms. $T_1$ and $T_2$ are augmentations on input data. The first term in Eq. 5 ensures that the generated perturbation has the property of promoting the similarity between two augmented views of the same sample to reduce $\mathcal{L}_{\mathrm{CL}}$. In this way, the perturbation provides easy-to-learn shortcut information for unsupervised learning such that the model will learn to extract the perturbation instead of the intrinsic semantic information in the data. In other words, the first term is to ensure the unlearnability against unsupervised training. The second term can enhance the linear separability of the perturbations. Furthermore, we propose to solve this bi-level optimization problem in Eq. 5 alternately:

$$\begin{cases} \mathrm{S1} : \theta^{(t)} = \arg\min_{\theta} \sum_{\boldsymbol{x}_i \in \mathcal{D}_c} \mathcal{L}_{\mathrm{CL}}\left(f\big(\theta, T_1(\boldsymbol{x}_i + \boldsymbol{\delta}_i^{(t-1)})\big), f\big(\theta, T_2(\boldsymbol{x}_i + \boldsymbol{\delta}_i^{(t-1)})\big)\right) \\ \mathrm{S2} : \boldsymbol{\delta}_i^{(t)} = \arg\min_{\{\boldsymbol{\delta}_i : \|\boldsymbol{\delta}_i\|_{\infty} \leq \epsilon\}} \mathcal{L}_{\mathrm{CL}}\left(f\big(\theta^{(t)}, T_1(\boldsymbol{x}_i + \boldsymbol{\delta}_i)\big), f\big(\theta^{(t)}, T_2(\boldsymbol{x}_i + \boldsymbol{\delta}_i)\big)\right) + \lambda \mathcal{L}_{\mathrm{CSD}}(\{\boldsymbol{\delta}_i, y_i\}_{i=1}^{n}). \end{cases} \quad (6)$$

In the first step (S1), we update the model parameter $\theta$ to minimize the unsupervised loss $\mathcal{L}_{\mathrm{CL}}$, while in the second step (S2) we optimize the perturbation $\{\boldsymbol{\delta}_i\}$ to jointly reduce the unsupervised loss and force the linear separability among different classes. By the bi-level optimization on unsupervised loss and CSD, we can generate UEs with both data-wise and training-wise transferability.

**Interpolation for data-wise transferability.** Once the generation stage is completed on the training dataset, $\mathcal{D}_c$, we can directly transfer the obtained perturbation to a new dataset, $\mathcal{D}_{\tilde{c}}$. Nevertheless, it is possible that the number of classes in $\mathcal{D}_c$ or the number of samples in one class in $\mathcal{D}_c$ is less than $\mathcal{D}_{\tilde{c}}$. The generated perturbation may not cover every sample in $\mathcal{D}_{\tilde{c}}$. To solve this problem, we use interpolation to create more perturbations. Interpolation can make use of current perturbation in $\Delta_{\mathcal{D}_c}$ to enlarge its size and transfer to a larger dataset. If more classes are required, we can interpolate between two current classes in $\mathcal{D}_c$ to create new classes:

$$\boldsymbol{\delta}_k^* = \alpha\boldsymbol{\delta}_i + (1 - \alpha)\boldsymbol{\delta}_j, \text{where } y_i \neq y_j. \quad (7)$$

If more samples (in one class) are required, we can interpolate within the class to create new samples:

$$\boldsymbol{\delta}_k^* = \alpha\boldsymbol{\delta}_i + (1 - \alpha)\boldsymbol{\delta}_j, \text{where } y_i = y_j. \quad (8)$$

By varying $\alpha$, more than one sample can be created from the interpolation between $\boldsymbol{\delta}_i$ and $\boldsymbol{\delta}_j$. Empirical results in Section 5.3 show that this interpolation strategy works very well.

## 5 EXPERIMENT

In this section, we validate the types of transferability of TUE. Section 5.1 introduces the experimental setups. Section 5.2 and Section 5.3 present the results on the improved training-wise and data-wise transferability, respectively. Section 5.4 illustrates the enhanced linear separability in input space by visualization. And Section 5.5 shows that TUE has both class-wise and sample-wise characteristics. Additional experiments can be found in Appendix C and Appendix D, which include visualization of interpolation, comparison with other methods, unlearnability on partly perturbed dataset and so on.

### 5.1 EXPERIMENTAL SETUPS

**Datasets.** The datasets include CIFAR-10 and CIFAR-100 (Krizhevsky et al., 2009), which contain 50,000 training images and 10,000 test images, and SVHN (Netzer et al., 2011), which contains 73,257 training images of ten classes and 26,032 test images. We randomly sample from SVHN to construct SVHN-small where the number of training images in every class is no more than 5000.

**Baselines.** We use three representative unlearnable examples as baselines, which are Error-Minimizing Noise (EMN) (Huang et al., 2020), Unlearnable Contrastive Learning (UCL) (He et al., 2022) and Synthetic Noise (SN) (He et al., 2022). More details can be found in Appendix B.1.

**Generation and Evaluation settings.** All the perturbations, except SN that is sampled from a distribution, are generated by PGD (Madry et al., 2018) on ResNet-18 and constrained by $\|\boldsymbol{\delta}_i\|_{\infty} \leq \epsilon$ with $\epsilon = 8/255$. In the evaluation stage, we use cross-entropy loss in supervised training and linear probing after contrastive pre-training in unsupervised training. The supervised model is trained for 200 epochs. The unsupervised model is pre-trained for 1000 epochs and fine-tuned for 100 epochs. More hyperparamaters for generation and evaluation can be found in Appendix B.2.

**Backbones.** We generated TUE on three unsupervised backbones, SimCLR (Chen et al., 2020a), MoCo (Chen et al., 2020b) and SimSiam (Chen & He, 2021). The backbones are also used in UCL. EMN is generated on the backbone architecture of ResNet-18 in a supervised setting. SN has no backbone in the generation stage and is directly sampled from the manually designed distribution.

Table 4: Test accuracy (%) of ResNet-18 trained on clean data vs. UEs by EMN, SN, UCL and TUE. TUE and UCL are generated on different unsupervised backbones. Under each backbone, the UEs are applied to the same unsupervised learning plus linear probing in the evaluation stage. The generation processes of EMN and SN are always the same regardless of evaluation and backbone of unsupervised learning. EMN and SN are also tested under the three backbones in the evaluation stage for comparison.

| Dataset | Backbone | Evaluation | Clean | EMN | SN | UCL | TUE |
|---------|----------|------------|-------|-----|-----|-----|-----|
| CIFAR-10 | SimCLR | Supervised | 93.79 | 14.74 | 19.23 | 92.86 | **10.67** |
| | | Unsupervised | 90.04 | 89.79 | 88.93 | **47.78** | 52.38 |
| | MoCo | Supervised | 93.79 | 14.74 | 19.23 | 92.62 | **10.06** |
| | | Unsupervised | 89.90 | 89.18 | 89.32 | **44.24** | 63.38 |
| | SimSiam | Supervised | 93.79 | 14.74 | 19.23 | 93.50 | **10.03** |
| | | Unsupervised | 90.59 | 91.43 | 91.54 | **30.43** | 35.57 |
| CIFAR-100 | SimCLR | Supervised | 74.49 | 5.23 | 2.13 | 72.17 | **0.76** |
| | | Unsupervised | 63.68 | 62.00 | 62.31 | **16.68** | 19.51 |
| | MoCo | Supervised | 74.49 | 5.23 | 2.13 | 71.59 | **1.09** |
| | | Unsupervised | 63.03 | 60.62 | 61.81 | **18.74** | 23.60 |
| | SimSiam | Supervised | 74.49 | 5.23 | 2.13 | 71.84 | **1.21** |
| | | Unsupervised | 64.69 | 65.96 | 66.83 | **4.64** | 6.17 |

## 5.2 TRAINING-WISE TRANSFERABLE UNLEARNABILITY

As mentioned above, existing UEs focus on one training setting, either supervised or unsupervised training. In this work, we use the proposed TUE to make the dataset unlearnable in both supervised and unsupervised training settings and thus protect the data from unauthorized training in a comprehensive way. In Table 4, we reported the comparison on CIFAR-10 and CIFAR-100 with supervised training by cross-entropy loss and three unsupervised training methods. The unlearnable examples are evaluated by two training settings, supervised and unsupervised training. When testing with unsupervised training, we use the corresponding unsupervised algorithm that is used to generate TUE.

Table 4 shows the model accuracy on clean test data after being trained on unlearnable data. First, we observe that existing unlearnability methods can work under one training setting, but have almost no protection under the other training setting. Specifically, EMN and SN reduces the test accuracy of supervised training to less than 20% on CIFAR-10 and less than 6% on CIFAR-100 but cannot transfer well in all of the three unsupervised trainings. UCL reduces the accuracy of unsupervised training to less than 50% on CIFAR-10 and less than 20% on CIFAR-100 but has almost no effect against supervised training. Second, TUE can hold the unlearnability in both supervised and unsupervised training settings. It reduces the supervised accuracy to around 10% on CIFAR-10 and 1% on CIFAR-100 and reduces the unsupervised accuracy to a comparable level to UCL. In particular, it can even perform better than EMN in supervised training. The test accuracy of unauthorized classifiers trained on TUE is around 4% lower than EMN on both CIFAR-10 and CIFAR-100. It indicates that the enhanced linear separability can mislead supervised models focusing more on TUE perturbation than EMN perturbation. In summary, from Table 4, we can see that TUE has good training-wise transferability, which can maintain the unlearnable effect when being transferred from supervised training to unsupervised training and it is the only one that maintains the unlearnable effects under two training settings.

## 5.3 DATA-WISE TRANSFERABLE UNLEARNABILITY

In this subsection, we show that TUE also has better data-wise transferability, i.e., the perturbations generated with TUE on one dataset can also protect any other datasets from unauthorized supervised training. Following the experimental settings in the preliminary studies of Section 3.2.2, we test the data-wise transferability first on non-target data samples, and then on non-target datasets. The results are reported in Table 5 and Table 6, respectively.

Table 5: Test accuracy (%) with different correpondings between train data and perturbations.

| Dataset | Methods | Original | Intra | Inter |
|---|---|---|---|---|
| CIFAR-10 | EMN | 15.88 | 30.74 | 33.91 |
| | SN | 14.07 | 13.59 | 13.15 |
| | TUE (SimCLR) | 10.67 | 10.16 | 10.90 |
| | TUE (MoCo) | 10.06 | 12.04 | 8.57 |
| | TUE (SimSiam) | 10.03 | 10.25 | 10.47 |
| CIFAR-100 | EMN | 6.59 | 21.63 | 35.50 |
| | SN | 2.13 | 2.44 | 2.73 |
| | TUE (SimCLR) | 0.76 | 1.11 | 1.13 |
| | TUE (MoCo) | 1.09 | 1.25 | 3.63 |
| | TUE (SimSiam) | 1.21 | 1.28 | 1.08 |

Table 6: Test accuracy (%) of transferring the perturbation generated on CIFAR-10 to different datasets.

| | SVHN-small | CIFAR-100 | SVHN |
|---|---|---|---|
| EMN | 27.59 | 21.80 | 24.72 |
| SN | 9.58 | 9.35 | 7.77 |
| TUE (SimCLR) | 9.77 | 10.53 | 11.72 |
| TUE (MoCo) | 11.29 | 8.32 | 13.95 |
| TUE (SimSiam) | 10.28 | 5.10 | 12.93 |

First, for non-target data samples, we compare the performance under 1) the original correspondence, 2) intra-class swapping, and 3) inter-class swapping between samples and perturbations in the same dataset. In Table 5, it is evident that our TUE can maintain high unlearnability (low test accuracy) under different correspondences on CIFAR-10 and CIFAR-100. Particularly, on CIFAR-100, the test accuracy of EMN under intra-class and inter-class swappings are roughly 15% and 29% higher than that in the original correspondence, but for TUE, the difference in the test accuracy under different correspondences is less than 3%. Although we can also notice this data-wise transferability in SN, it is not an optimizable method and is unable to be used for improving the training-wise transferability.

Second, to verify data-wise transferability on non-target datasets, we generate TUE on CIFAR-10 and test the performance of transferring onto three datasets, SVHN-small, CIFAR-100 and SVHN. TUE on CIFAR-10 can be directly transferred to SVHN-small without interpolation, while CIFAR-100 requires interpolation for more classes and SVHN requires interpolation for more samples. The baseline methods are also interpolated in the same way for comparison. In Table 6, we can observe that on all the datasets, TUE can maintain the unlearnable effect onto all the other datasets. The enhanced linear separability of TUE ensures that the test accuracy is low. For example, EMN has poorer protection after transferred onto SVHN-small with the accuracy of 27.59%, but TUE can maintain the unlearnability on non-target datasets for all the backbones at around 10% accuracy.

From Table 5 and Table 6, we can draw the conclusion that the proposed TUE has better data-wise transferability than optimization-based unlearnable methods like EMN, and our strategy to transfer TUE to protect other datasets can be very efficient.

## 5.4 LINEAR SEPARABILITY OF TRANSFERABLE UNLEARNABLE EXAMPLES

In this subsection, we visualize the clusters formed by different types of UEs to gain deeper understandings of how linear separability influences unlearnability. Particularly, we use t-SNE (Van der Maaten & Hinton, 2008) to show the perturbation in the input space. Although t-SNE cannot accurately describe the high-dimension space, it is useful to observe the separability of the perturbations in the input space. We compare the perturbations in the input space in the following three groups.

**Comparison of Linear separability in supervised unlearnable examples.** In Fig. 4, we show the perturbations of EMN, SN and TUE to understand why linear separability is more effective than EMN when used in supervised classification. In Fig. 4, the perturbations of different classes in TUE and SN are clearly separated and have no overlapping. In contrast, EMN has some data points mixed with other classes which are confusing when used in classification. Meanwhile the clusters in EMN do

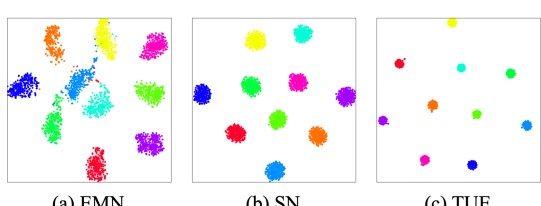

| (a) EMN | (b) SN | (c) TUE |
|---|---|---|

Figure 4: t-SNE visualization of linear separability of supervised unlearnable examples.

not contract as well as TUE and SN. Since better linear separability can provide useful information about classification, the supervised model will focus on TUE and SN more than EMN and ignore the semantic features in clean images.

**Linear separability in Interpolation.** In Fig. 5, we show that the interpolation on TUE from CIFAR-10 to SVHN and CIFAR-100 can also hold linear separability which makes it data-wise transferable. We can observe that in the interpolation for SVHN, although the size of each class is different,

they can still hold clear linear separability and unlearnbility. In SVHN, the interpolation within classes creates more samples for one class. Since the numbers of examples in different classes in SVHN are different, it can be observed that some clusters are much larger than others. For CIFAR-100, more new classes are created and they still keep good linear separability. So the supervised model trained on interpolation-based unlearnable SVHN and CIFAR-100 will learn nothing but the perturbations and perform poorly on clean test data which has no perturbations.

**Comparison of Linear separability between supervised and unsupervised unlearnability.** In Fig. 6, we show the perturbations of UCL and TUE in input space to understand why UCL cannot help with unlearnable effects in supervised training. TUE is linear separable, which can be used to protect the data from both supervised and unsupervised training. But for UCL, all the perturbations are mixed, they cannot provide the class information for supervised training. So for a supervised model, UCL is not a helpful feature and will not vanquish the semantic feature.

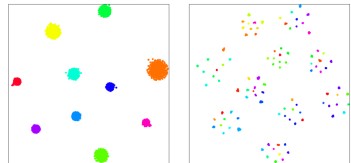

(a) Interpolation: SVHN  (b) Interpolation: CIFAR-100

Figure 5: t-SNE visualization of linear separability of interpolations.

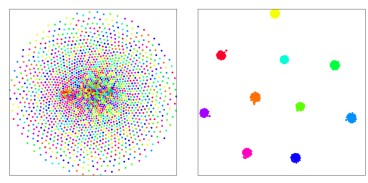

(a) UCL          (b) TUE

Figure 6: t-SNE visualization of linear separability of supervised and unsupervised unlearnability.

## 5.5 CASE STUDY

In this subsection, by showing the patterns of the perturbations generated by different methods, we can deepen our understanding on linear separability in TUE and observe the classwise and samplewise characteristics, which reflect the basic idea behind training-wise transferability.

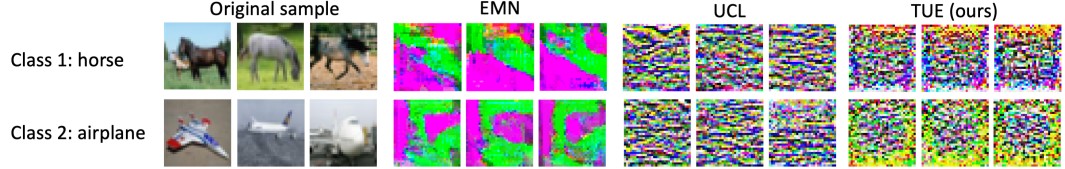

Figure 7: Visualization of the unlearnable examples of EMN, UCL and TUE(ours). We rescale the perturbation from [-8/255, 8/255] to [0,1] for better visualization.

We illustrate the samples and unlearnable perturbations from two classes of CIFAR-10, i.e., horse and airplane, in Figure 7. First, we find that EMN and TUE both have class-wise noise patterns. Every noisy sample has consistent patterns in one class, while every class has a different feature pattern from other classes. This is a new perspective to show that TUE has linear separability and perturbations from the same class are clustered in input space. Second, in UCL, the perturbations can be more diverse and related to each sample, which indicates that unsupervised unlearnability requires more complex feature patterns. We cannot see the class-wise patterns like EMN. Because UCL uses no label information and has no concept of classes. Third, we notice that TUE has both class-wise and sample-wise characteristics as expected. The class-wise feature can be the easy-to-learn feature for the supervised model, while the samplewise feature can be effective in reducing unsupervised loss, which makes the unlearnability transferable from supervised training to unsupervised training. In summary, from the case study, we can observe that the co-existing class-wise and sample-wise features provide supervised and unsupervised protection at the same time. The linear separability makes the supervised unlearnability data-wise transferable.

## 6 CONCLUSION

We reveal the limitation of training-wise and data-wise transferability in existing UEs and propose CSD to enhance transferability. The proposed Transferable Unlearnable Examples has training-wise transferability and can protect data from unauthorized usage in a comprehensive way. Meanwhile, TUE has data-wise transferability and can generate unlearnbility once for all, which makes UEs more practical and efficient. TUE greatly pushes the boundary of existing unlearnable methods that can only work on the target data and target training setting. We improve the data-wise and training-wise transferability of unlearnable examples and provide more flexible, practical, and comprehensive protections from unauthorized data usage.

## ACKNOWLEDGMENTS

Jie Ren, Han Xu, Yuxuan Wan and Jiliang Tang are supported by the National Science Foundation (NSF) under grant numbers CNS1815636, IIS1845081, IIS1928278, IIS1955285, IIS2212032, IIS2212144, IOS2107215, and IOS2035472, the Army Research Office (ARO) under grant number W911NF-21-1-0198, the Home Depot, Cisco Systems Inc, Amazon Faculty Award, JohnsonJohnson and SNAP. Xingjun Ma is in part supported by the National Key R&D Program of China (Grant No. 2021ZD0112804), the National Natural Science Foundation of China (Grant No. 62276067), and the Science and Technology Commission of Shanghai Municipality (Grant No. 22511106102).

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

## A EXISTING UNLEARNABLE METHODS CANNOT PROTECT UNLABELED DATASET IN A TRAINING-WISE TRASNFERABLE WAY

Previous unlearnable methods can only protect data from one unauthorized training setting. According to whether the dataset is annotated, unlearnable examples can be classified in two settings, supervised and unsupervised unlearnable examples, as mentioned in Section 2. The supervised unlearnable methods, like EMN, can create unlearability based on the label information and protect the data from supervised training, but these methods cannot prevent the unauthorized parties from training the data with an unsupervised algorithm. Similarly, the unsupervised unlearnable examples produced by Unleanrable Contrastive Learning (UCL) He et al. (2022) can destroy the unsupervised training but they are still learnable in the supervised training. In Fig. 8, EMN decreases the accuracy of supervised training from 93.8% to 14.7%, but only decreases the accuracy of unsupervised training by 0.2%. Similarly, SN decreases supervised training to 14.1%, but only decreases unsupervised training by 1.1%. Supervised unlearnable examples have almost no training-wise transferability to unsupervised training. UCL, which produces unsupervised unlearnable examples, can protect data from unsupervised training, but cannot be transferred well to supervised training. Under unsupervised training, UCL can decrease the test accuracy from 90.0% to 47.8%, but under supervised training, it can only decrease the accuracy by 0.9%.

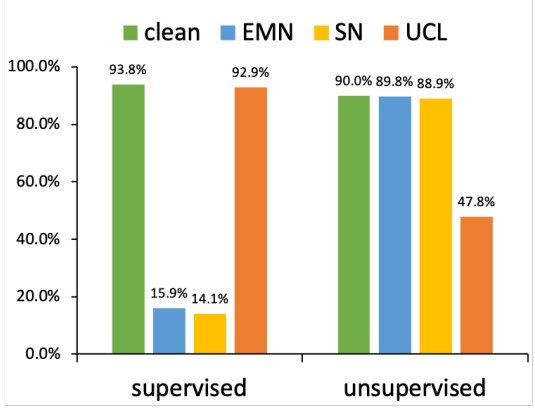

Figure 8: Accuracy on test data of classifiers produced by supervised training and unsupervised training on the unlearnable examples from CIFAR-10

## B  DETAILS OF EXPERIMENTAL SETTINGS

### B.1  DETAILS OF BASELINE METHODS

**EMN** (Huang et al., 2020) generates the easy-to-learn feature by inducing the perturbations to reduce the supervised cross-entropy loss via alternating optimization on the bi-level min-min problem:

$$\min_{\theta} \min_{\boldsymbol{\delta}_i \in \{\boldsymbol{v}:\|\boldsymbol{v}\|_\infty \leq \epsilon\}} \sum_{i=1}^{n} \mathcal{L}\left(f_\theta\left(\boldsymbol{x}_i + \boldsymbol{\delta}_i\right), y_i\right). \tag{9}$$

The outer minimization can imitate the training process, while the inner minimization can induce $\boldsymbol{\delta}_i$ to have the property of minimizing the supervised loss. Due to this property, deep models will pay more attention to the easy-to-learn $\boldsymbol{\delta}_i$ and ignore $\boldsymbol{x}_i$.

**UCL** (He et al., 2022) also generates perturbatioins which is easy-to-learn for unsupervised contrastive learning by inducing the perturbations to reduce the contrastive loss via alternating optimization on the bi-level min-min problem:

$$\min_{\theta} \min_{\{\boldsymbol{\delta}_i:\|\boldsymbol{\delta}_i\|_\infty \leq \epsilon\}} \sum_{i=1}^{n} \mathcal{L}_{\text{CL}}\left(f\left(\theta, T_1(\boldsymbol{x}_i + \boldsymbol{\delta}_i)\right), f\left(\theta, T_2(\boldsymbol{x}_i + \boldsymbol{\delta}_i)\right)\right). \tag{10}$$

It does not require label information and works well on different contrastive learning backbones.

**SN** (Yu et al., 2021) is sampled from a manually designed distribution. It is not targeting any backbone or dataset and only focuses on linear separability. SN first sets $M$ centers in the perturbation space, where $M$ is the number of protected classes. The generation process can be seen as sampling the perturbations for one class from a Gaussian distribution whose mean is one of the $M$ centers.

### B.2  SETTINGS OF GENERATION PROCESS

All the baselines and our proposed TUE are generated in the way of PGD Madry et al. (2018), except for SN, which is sampled from a manually designed distribution following the algorithm in Yu et al. (2021). For EMN, UCL, and TUE, the generation process is an alternate optimization between model parameters and perturbations. For example, TUE is optimized in the way of Eq. 6. In EMN, after every epoch of optimization on model parameters, the whole set perturbations are optimized for one epoch by PGD-20. In TUE, different backbones are set to different schedule. For TUE (SimCLR), the model parameters are trained for 40 epochs, and after every 1/5 epoch of optimization on model parameters, the whole set perturbations are optimized for one epoch by PGD-20. For TUE (MoCo), the model parameters are trained for 200 epochs, and after every epoch of optimization on model parameters, the whole set perturbations are optimized for one epoch by PGD-5. For TUE (SimSiam), the model parameters are trained for 50 epochs, and after every 1/4 epoch of optimization on model parameters, the whole set perturbations are optimized for one epoch by PGD-20. UCL with SimCLR and SimSiam have the same settings as TUE. UCL with MoCo uses PGD-10 and the other settings in the schedule is the same as TUE (MoCo). Finally, all the perturbations are constrained by $l_\infty$ norm, i.e. $\|\delta_i\|_\infty \leq \epsilon$ where $\epsilon = 8/255$.

### B.3  SETTINGS OF EVALUATION STAGE

To evaluate the unlearnable effect, we use CrossEntropy as the objective function in the unauthorized supervised training. We use linear probing after contrastive pre-training to evaluate the unlearnable effect in unsupervised training. The supervised model is trained for 200 epochs. The unsupervised model is pre-trained for 1000 epochs by unsupervised contrastive learning and then fine-tuned for 100 epochs by linear probing. The details for the hyperparameters are listed in Table 7.

## C  VISUALIZATION OF INTERPOLATION

We use t-SNE to show the interpolations in the perturbation space (Fig. 9) and then visualize the perturbation patterns (Fig. 10). In Fig. 9, it shows that intra-class interpolation tends to create new perturbations around the original perturbation clusters, while inter-class interpolation tends

Table 7: Hyperparameters of evaluation stage

| Hyperparameter | Supervised | Unsupervised pretraining | | | Linear probing | | |
|---|---|---|---|---|---|---|---|
| | | SimCLR | MoCo | SimSiam | SimCLR | MoCo | SimSiam |
| Epoch | 200 | 1000 | 1000 | 1000 | 100 | 100 | 100 |
| Optimizer | SGD | SGD | Adam | SGD | Adam | SGD | SGD |
| Learning Rate | 0.1 | 0.06 | 0.3 | 0.06 | 0.001 | 30 | 30 |
| LR Scheduler | CosineAnnealingLR (CA) | - | CA | CA | - | CA | CA |
| Encoder Momentum | - | - | 0.99 | - | - | - | - |
| Loss Function | CrossEntropy (CE) | InfoNCE | InfoNCE | Similarity between positive samples | CE | CE | CE |

to create more separated clusters, extending the unlearnable effect to more classes. In Fig. 10, it shows interpolation within classes can keep the class-wise pattern that is the same as the source perturbations, while interpolation between classes can create a new class pattern. Both the linear separable clusters and the class-wise patterns can show that interpolation can keep good linear separability. By visualization we show the reason why interpolation can be an effective way for increasing the number of perturbations.

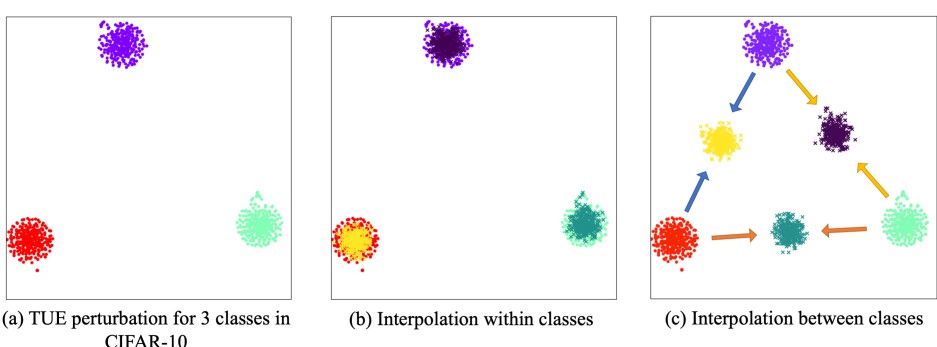

(a) TUE perturbation for 3 classes in CIFAR-10

(b) Interpolation within classes

(c) Interpolation between classes

Figure 9: t-SNE visualization of interpolation in perturbation space

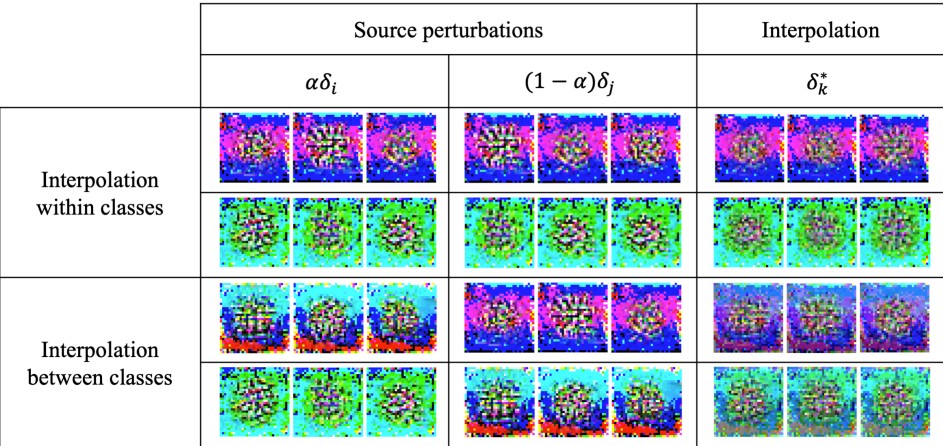

Figure 10: Visualization of patterns of interpolation

# D ADDITIONAL EXPERIMENTS

## D.1 COMPARISON WITH SIMPLY COMBINING EMN AND UCL

In this subsection, we want to compare TUE with simply combining the loss of EMN and UCL (EMN-UCL). EMN-UCL is a simple and direct method, but it will lose a part of the linear separability

of the perturbations, and thus reduce the data-wise transferability and protection against supervised training as shown in the experiments in Table 8. As we analyzed in Section 3.2.2 and Section 5.3, EMN is designed to reduce the supervised loss on target data, while TUE creates the shortcut by linear separability of perturbations, which is not tied to the specific data. As such, EMN-UCL should have limited data-wise transferability compared with TUE.

Table 8: Comparison between EMN-UCL and TUE.

| Backbone | Method | Training-wise transferability | | Data-wise transferability | | | |
| | | Supervised | Unsupervised | Intra-class | Inter-class | CIFAR-10 → SVHN | CIFAR-10 → CIFAR-100 |
|---|---|---|---|---|---|---|---|
| SimCLR | EMN-UCL | 19.89 | 47.83 | 16.15 | 14.82 | 79.25 | 45.61 |
| | TUE | 10.67 | 52.38 | 10.16 | 10.90 | 11.72 | 10.53 |
| MoCo | EMN-UCL | 13.34 | 50.62 | 13.61 | 11.90 | 28.52 | 27.43 |
| | TUE | 10.06 | 63.38 | 12.04 | 8.57 | 13.95 | 8.32 |
| SimSiam | EMN-UCL | 15.36 | 32.74 | 16.15 | 14.82 | 11.14 | 21.47 |
| | TUE | 10.03 | 35.57 | 10.25 | 10.47 | 12.93 | 5.10 |

In Table 8, the experimental results are given to show the following weaknesses of EMN-UCL:

- First, EMN-UCL has limited data-wise transferability, especially when being transferred onto non-target datasets. Because of the poor linear separability from EMN, the accuracies of EMN-UCL on non-target datasets are much higher on almost every dataset and backbone. EMN-UCL with SimCLR backbone even has an accuracy of 79.25% when transferred from CIFAR-10 to SVHN and 45.61% when transferred from CIFAR-10 to CIFAR-100. However, TUE can reduce the accuracy on SVHN to around 12% and CIFAR-100 to less than 11% on every backbone.

- Second, EMN-UCL has less protection against supervised training than TUE. On CIFAR-10, TUE can reduce the test accuracy of unauthorized supervised training to around 10%, while EMN-UCL has a higher accuracy which is from 13% to 19 % on different backbones. This means EMN-UCL provides less protection against supervised training than TUE.

In conclusion, TUE can provide better supervised unlearnability and better data-wise transferability than combining EMN and UCL.

## D.2 THE CHANGES OF CSD AFTER INTERPOLATION

To better understand how interpolation affects the linear separability, we show the change of linear separability after the interpolation in Table 9. The interpolation from CIFAR10 to SVHN has almost no change to CSD. Although the interpolation from CIFAR10 to CIFAR100 increases CSD a little, it is still low enough for good linear separability. The change for SN and TUE in CIFAR100 can be almost ignored. The reason why CSD increases in CIFAR100 is that a larger number of classes would make the perturbation space more crowded.

Table 9: The changes of CSD values after interpolation.

| | CIFAR-10 | CIFAR-10 to SVHN | CIFAR-10 to CIFAR-100 |
|---|---|---|---|
| EMN | 1.74 | 1.60 | 2.06 |
| SN | 0.55 | 0.61 | 0.66 |
| TUE (SimCLR) | 0.71 | 0.72 | 0.85 |
| TUE (MoCo) | 0.52 | 0.58 | 0.62 |
| TUE (SimSiam) | 0.67 | 0.69 | 0.79 |

## D.3 ABLATION STUDY

In this subsection, we conduct ablation experiments on the two terms of the proposed loss. The results are reported in Tables 10 and 11, which show that removing either term will greatly hurt the performance of TUE.

Particularly, in Table 10, we show that, if we only use CSD loss, the protection against unsupervised training will fail. In Table 11, we show that if only the Contrastive-Learning term is used (reduces to the UCL baseline), it will lose its protection against supervised training.

Table 10: The changes of CSD values after interpolation.

|  |  | Supervised | Unsupervised | | |
|---|---|---|---|---|---|
|  |  |  | SimCLR | MoCo | SimSiam |
| CIFAR-10 | Clean | 93.79 | 90.04 | 89.90 | 90.59 |
|  | CSD | 9.99 | 80.33 | 79.24 | 81.20 |
| CIFAR-100 | Clean | 74.49 | 63.68 | 63.03 | 64.69 |
|  | CSD | 1.03 | 55.69 | 51.81 | 58.99 |

Table 11: The changes of CSD values after interpolation.

| Backbone | Method | CIFAR-10 | | cifar-100 | |
|---|---|---|---|---|---|
|  |  | Supervised | Unsupervised | Supervised | Unsupervised |
| SimCLR | Clean | 93.79 | 90.04 | 74.49 | 63.68 |
|  | UCL | 92.86 | 47.78 | 72.17 | 16.68 |
| MoCo | Clean | 93.79 | 89.90 | 74.49 | 63.03 |
|  | UCL | 92.62 | 44.24 | 71.59 | 18.74 |
| SimCLR | Clean | 93.79 | 90.59 | 74.49 | 64.69 |
|  | UCL | 93.50 | 30.43 | 71.84 | 4.64 |

## D.4 COMPARISON WITH SUPERVISED CONTRASTIVE LEARNING

The loss of Supervised Contrastive Learning (SCL) has a similar intuition to our CSD. SCL pulls samples within the same class to be closer, while pushing those from different classes far away. However, it is not a suitable loss for unlearnable examples.

First, SCL cannot quantify linear separability, which is the key property for achieving training-wise and data-wise transferability. Particularly, CSD uses the radio between the intra-class and inter-class distances to measure the overlap between two classes. But SCL takes all the samples that belong to other classes as negative which can not reflect the distance between two classes.

Second, SCL performs worse than CSD. In Table 12, we test the unlearnable effect of SCL and our TUE with MoCo. It shows that SCL works against supervised exploitation, but it is less effective in preventing unsupervised exploitation.

Table 12: Comparison between SCL and TUE.

|  |  | Supervised | Unsupervised |
|---|---|---|---|
| CIFAR-10 | SCL(MoCo) | 10.12 | 66.46 |
|  | TUE(MoCo) | 10.06 | 63.38 |
| CIFAR-100 | SCL(MoCo) | 1.29 | 28.11 |
|  | TUE(MoCo) | 1.09 | 23.60 |

In conclusion, although SCL is similar to CSD in the idea of pushing clusters, CSD is more suitable to TUE because it can quantify the linear separability and performs better in the protection against unsupervised training.

## D.5 PERTURBING A PART OF CLASSES IN THE DATASET

In some cases, we have access to the whole dataset. Taking social media platforms as an example, for the company, they can protect all the data released on the social media and they do have access to all the data. For users, they can protect their own data by perturbing before uploading.

In this paper, we focus on the cases where the protection can be added onto all the data. However, we also evaluate when people can only perturb part of the classes of the data (like the users can only perturb their own data) by the experiments shown in Table 13. From Table 13, we can see that although the unsupervised training can still learn semantic features from the partly perturbed data, partly perturbing the dataset can protect data from supervised training very effectively, the accuracy

on perturbed classes is almost 0 for most cases. When we only have access to a part of the data, the unlearnable examples can still protect that part of classes.

Table 13: Test accuracy (%) on partly perturb datasets. The accuracy is calculated on the perturbed classes.

| | CIFAR-10 | | | | cifar-100 | | | |
|---|---|---|---|---|---|---|---|---|
| Number of perturbed classes | 1 | | 2 | | 1 | | 10 | |
| Unauthorized training | Supervised | Unsupervised | Supervised | Unsupervised | Supervised | Unsupervised | Supervised | Unsupervised |
| TUE(SimCLR) | 0.20 | 93.40 | 0.15 | 92.58 | 4.00 | 88.00 | 0.90 | 69.00 |
| TUE(MoCo) | 0.00 | 94.40 | 0.05 | 95.55 | 4.00 | 83.30 | 0.40 | 73.30 |
| TUE(SimSiam) | 0.10 | 82.70 | 0.05 | 86.15 | 7.00 | 83.00 | 0.20 | 55.60 |

## D.6 UNLEARNABLE EFFECT OF NEW CLASSES FROM INTERPOLATION

In this subsection, we show that the new classes of perturbations created by interpolation have good unlearnable effect and data-wise transferability. As shown in Sections 4 and 5.3, we used interpolation to create more classes of perturbations to transfer TUE from CIFAR10 to CIFAR100, i.e., 90 new classes of perturbations are created to protect CIFAR100, while the remaining 10 classes are directly transferred from CIFAR10. To better understand how the unlearnable effect on interpolated new classes is, we show it from two perspectives, i.e., the unlearnable effect against unauthorized training and linear separability (quantified by CSD). First, in Table 14, we show that the unlearnable effect on the new created classes is as good as those directly transferred from CIFAR-10 to CIFAR-100. Second, in Table 15, we show that the linear separability of interpolated perturbations for new classes is almost the same as the linear separability for overall classes. In conclusion, the interpolated new classes have good linear separability and the unlearnable effect can transfer well on non-target dataset.

Table 14: Test accuracy (%) on new classes from interpolation.

| From CIFAR-10 to CIFAR-100 | New classes | Overall classes |
|---|---|---|
| TUE(SimCLR) | 6.41 | 10.53 |
| TUE(MoCo) | 7.35 | 8.32 |
| TUE(SimSiam) | 11.94 | 5.10 |

Table 15: CSD on new classes from interpolation.

| From CIFAR-10 to CIFAR-100 | New classes | Overall classes |
|---|---|---|
| TUE(SimCLR) | 0.8527 | 0.8507 |
| TUE(MoCo) | 0.6281 | 0.6265 |
| TUE(SimSiam) | 0.7979 | 0.7970 |

