# OpenReview forum: "Transferable Unlearnable Examples"
_ICLR.cc/2023/Conference — ICLR 2023 poster_

### Official Review · Reviewer_sZJu · 2022-10-26

**Confidence:** 4
**Correctness:** 3
**Technical Novelty And Significance:** 3
**Empirical Novelty And Significance:** 4
**Recommendation:** 6

**Clarity, Quality, Novelty And Reproducibility:**

The paper is well written and novel. The experimental settings are detailed for reproducibility



**Strength And Weaknesses:**

Strengths
-the paper proposes training-wise and data-wise transferability for unlearnable effects using contrastive learning
- the proposed methods uses linear separability in addition to similarity between augmentations
-experiments are extensive and detailed and show the improvement in data-wise and training-wise transferability



**Summary Of The Paper:**

The paper proposes Classwise Separability Discriminant (CSD), an unlearnable strategy to transfer the unlearnable effects (perturbations to avoid unauthorized data usage) to other training settings and datasets by enhancing the linear separability. Experiments showcase transferability of the proposed unlearnable examples across training settings and datasets

**Summary Of The Review:**

The paper addresses an important problem of improving data-wise and training-wise transferability of unlearnable examples. The proposed approach is novel and detailed results show it outperforms its counterparts.

---

> ### Author Response · Authors · 2022-11-07
> **Response to Reviewer sZJu**
>
> Thanks for your positive comments. As you mentioned, we identified the problem that previous unlearnable methods could not perform well in terms of training-wise and data-wise transferability, which limit their practical usage in real-world scenarios. To solve the problem, we proposed a novel method that exploits the advantages of linear separability and unsupervised contrastive learning to provide practical and comprehensive protection against the unauthorized exploitation of data. We thank the reviewer for highlighting our strengths, they are very encouraging. However, we did not notice any concerns or suggestions for our paper. It seems that the reviewer might have missed some of the comments in the "Strength And Weaknesses". Please kindly let us know if you have more questions.

---

### Official Review · Reviewer_5F6K · 2022-10-29

**Confidence:** 4
**Correctness:** 3
**Technical Novelty And Significance:** 2
**Empirical Novelty And Significance:** 2
**Recommendation:** 6

**Clarity, Quality, Novelty And Reproducibility:**

The paper is well-written. It identifies the limitation of supervised learnable examples and combines it with an unsupervised representation approach to make it data-wise and training-wise transferable. Novelty of the paper is very incremental, and the intended use case is somewhat clear. Experimental set-up is sound and results are reproducible.

**Strength And Weaknesses:**

- The paper systematically establishes the limitations of existing supervised unlearnable methods, and then proposes a simple strategy to make the unlearnable examples transfer to unsupervised and other datasets.

- The proposed strategy is simple, and adds a class-wise separability discriminant loss to the contrastive loss for learning a transferable representation.

- Interpolation data-wise transferability strategy is not convincing. Just linearly interpolating class samples or creating new class categories is not going to be representative of the real-world situations. Further experimentation is required to establish the claims to this end.

- Comparison with supervised contrastive learning that makes use of samples within the class categories as positives, and other class categories as negatives would be useful, as supervised contrastive learning yields representations that are similar to the ones desired from class separability discriminant.

- It is also not clear to me the rationale of the paper. If the goal is to avoid unauthorized use of the data, the data could be encrypted, or the security authentication protocol can be defined accordingly. Adding a noise to make it linearly separable seems difficult for challenging datasets.


**Summary Of The Paper:**

This paper presents an unlearnable strategy to prevent third parties from training on the data without authorization. The unlearnable strategy is based on class-wise separability discriminant that seeks to transfer better to other training settings and datasets, in contrast to existing methods that only work on a target training setting and a target dataset. Experiments on digit datasets suggest the feasibility of generating unlearnable examples across training settings and datasets.

**Summary Of The Review:**

The paper presents an application of representation learning to dataset protection and security. It identifies the limitation of supervised unlearnable examples in transferring to a new situation, and presents a remedy to make unlearnable examples transferable. More experiments on use cases and challenging datasets will improve the quality of the paper.

---

> ### Author Response · Authors · 2022-11-19
> **Response to Reviewer 5F6K (1/3)**
>
> Thank you very much for the valuable feedback. Please find our response below for your concerns.
>
> > **Q1** It is also not clear to me the rationale of the paper. If the goal is to avoid unauthorized use of the data, the data could be encrypted, or the security authentication protocol can be defined accordingly. Adding a noise to make it linearly separable seems difficult for challenging datasets.
>
> **A1** Thanks for your question about unlearnable examples. This is a problem that is well studied like in [1, 2, 3, 4, 5]. In certain scenarios, people want to publish their data for others to read, watch and browse, but they do not want others to collect and exploit their data to train commercial or malicious machine learning models. For example, in social media platforms, people usually share their selfies to the public. But they don’t want unauthorized people to use that data to train a facial recognition model, which is risky. The unlearnable example is a different solution from encryption. Because encryption will make the data directly inaccessible for others. However, unlearnable examples can maintain the normal usage, and prevent unauthorized training.
>
> [1] Huang, Hanxun, et al. "Unlearnable Examples: Making Personal Data Unexploitable." International Conference on Learning Representations. 2021.
>
> [2] Fowl, Liam H., et al. "Adversarial Examples Make Strong Poisons." Advances in Neural Information Processing Systems. 2021.
>
> [3] Yu, Da, et al. "Availability attacks create shortcuts." Proceedings of the 28th ACM SIGKDD Conference on Knowledge Discovery and Data Mining. 2022.
>
> [4] Fu, Shaopeng, et al. "Robust unlearnable examples: Protecting data privacy against adversarial learning." International Conference on Learning Representations. 2022.
>
> [5] He, Hao, Kaiwen Zha, and Dina Katabi. "Indiscriminate Poisoning Attacks on Unsupervised Contrastive Learning." 2022.

---

> > ### Author Response · Authors · 2022-11-19
> > **Response to Reviewer 5F6K (2/3)**
> >
> > > **Q2** Interpolation data-wise transferability strategy is not convincing. Just linearly interpolating class samples or creating new class categories is not going to be representative of the real-world situations. Further experimentation is required to establish the claims to this end.
> >
> > **A2** Thanks for your comment about interpolation. Please first allow us to kindly clarify  the typos in the Eq. 7 and Eq. 8 for interpolation. This may cause you to think that the interpolation is conducted between two samples. The interpolation is conducted between two (unlearnable) perturbations, rather than two samples. Specifically, in Eq. 7, $\boldsymbol{x}$ should be corrected to $\boldsymbol{\delta}$, and the correct version of Eq. 7 should be
> > $$
> > \boldsymbol{\delta}_k^*=\alpha \boldsymbol{\delta}_i+(1-\alpha) \boldsymbol{\delta}_j \text {, where } y_i \neq y_j
> > $$
> > Eq. 8 should also be corrected as
> > $$
> > \boldsymbol{\delta}_k^*=\alpha \boldsymbol{\delta}_i+(1-\alpha) \boldsymbol{\delta}_j, \text { where } y_i=y_j
> > $$
> > With this correction, we can continue to explain the interpolation clearly.
> >
> > By interpolating the perturbations, we can transfer them to create more unlearnable examples for larger datasets (those with more classes, or the same classes but more samples). The main reason why the interpolated perturbations can work is that they still hold good linear separability after interpolation. As we demonstrated in Section 3.2, linear separability leads to easy-to-learn features that can trick the model to ignore the real contents. To better understand the interpolation mechanism, we show how it affects the distribution of the perturbations in Figures 9 and 10 in Appendix C, and how it affects the linear separability via CSD in Table 9 in Appendix D.2.
> >
> > - In Figures 9 and 10, we use t-SNE to show the interpolations in the perturbation space (Figure 9) and then visualize the perturbation patterns (Figure 10). In Figure 9, it shows that intra-class interpolation tends to create new perturbations around the original perturbation clusters, while inter-class interpolation tends to create more separated clusters, extending the unlearnable effect to more classes. In Figure 10, it shows interpolation within classes can keep the class-wise pattern that is the same as the source perturbations, while interpolation between classes can create a new class pattern. Both the linear separable clusters and the class-wise patterns can show that interpolation can keep good linear separability. By visualization we show why interpolation can be an effective way for perturbations.
> >
> > - From Table 9, the interpolation mechanism can also be understood from the perspective of CSD. As shown in Table 9, the interpolation from CIFAR10 to SVHN has almost no change to CSD. Although the interpolation from CIFAR10 to CIFAR100 increases CSD a little, it is still low enough for good linear separability. The change for SN and TUE in CIFAR100 can be almost ignored. This means interpolation has almost no impact on linear separability. (The reason why CSD increases in CIFAR100 is that a larger number of classes would make the perturbation space more crowded.)
> >
> >   [*Table* 9: The change of the CSD value after interpolation]
> >
> >   |              | CIFAR10 | CIFAR10 -> SVHN | CIFAR10 -> CIFAR100 |
> >   | ------------ | ------- | --------------- | ------------------- |
> >   | EMN          | 1.7404  | 1.6066          | 2.0615              |
> >   | SN           | 0.5584  | 0.6155          | 0.6622              |
> >   | TUE(SimCLR)  | 0.7149  | 0.7298          | 0.8507              |
> >   | TUE(MoCo)    | 0.5276  | 0.5842          | 0.6265              |
> >   | TUE(SimSiam) | 0.6704  | 0.6921          | 0.797               |
> >
> > In conclusion, interpolation between perturbations is an effective way to transfer the unlearnable perturbations onto larger datasets.

---

> > > ### Author Response · Authors · 2022-11-19
> > > **Response to Reviewer 5F6K (3/3)**
> > >
> > > > **Q3** Comparison with supervised contrastive learning that makes use of samples within the class categories as positives, and other class categories as negatives would be useful, as supervised contrastive learning yields representations that are similar to the ones desired from class separability discriminant.
> > >
> > > **A3** Thanks for your insightful suggestions. As suggested, we make the comparison between supervised contrastive learning (SCL) and CSD. The loss of supervised contrastive learning (SCL) pulls samples within the same class to be closer, while pushing those from different classes far away, which is similar to our CSD. However, it is not a suitable loss for unlearnable examples.
> > >
> > > - First, SCL cannot quantify linear separability, which is the key property for achieving training-wise and data-wise transferability. Particularly, CSD uses the radio between the intra-class and inter-class distances to measure the overlap between two classes. But SCL takes all the samples that belong to other classes as negative which can not reflect the distance between two classes.
> > > - Second, SCL performs worse than CSD. In Table 12 in Appendix D.4, we test the unlearnable effect of SCL and our TUE with MoCo. It shows that SCL works against supervised exploitation, but it is less effective in preventing unsupervised exploitation.
> > >
> > > [*Table 12*: Unlearnable effect of SCL and TUE]
> > > |          |           | supervised | unsupervised |
> > > | -------- | --------- | ---------- | ------------ |
> > > | CIFAR10  | SCL(MoCo) | 10.12      | 66.46        |
> > > |          | TUE(MoCo) | 10.06      | 63.38        |
> > > | CIFAR100 | SCL(MoCo) | 1.29       | 28.11        |
> > > |          | TUE(MoCo) | 1.09       | 23.6         |
> > >
> > > In conclusion, although SCL is similar to CSD in the idea of pushing clusters, CSD is more suitable to TUE because it can quantify the linear separability and performs better in the protection against unsupervised training.

---

### Official Review · Reviewer_esp8 · 2022-11-01

**Confidence:** 4
**Correctness:** 3
**Technical Novelty And Significance:** 3
**Empirical Novelty And Significance:** 3
**Recommendation:** 8

**Clarity, Quality, Novelty And Reproducibility:**

Clarity
The algorithm of this paper is clarity and easy to understand.
However, something is puzzling in experimental results.

Quality
This paper provides an effective strategy and some meaningful conclusions.

Novelty
The contributions are significant and somewhat new. Aspects of the contributions exist in prior work.



**Strength And Weaknesses:**

Strength:
1. This paper proposed a simple but effective unlearnable strategy that can preserve both training-wise transferability and data-wise transferability.
2. The paper is easy to understand, the formula is concise, and the algorithm description is clear.
3. The performance gain seems to be significant.
4. The visualization of unlearnable examples helps understand the effectiveness of linear separability.

Weaknesses:
1. In section 3, the authors mentioned that this paper focuses on supervised unlearnable examples, which seems to contradict the training-wise transferability of the proposed method.
2. The first term of equation 5 combines unsupervised loss explicitly, while the second term L_S combines supervised loss implicitly. What if we consider both unsupervised and supervised loss explicitly?
3. It is not clear how much interpolation affects data-wise transferability.
4. What are the backbones of EMN and SN in Table 5 and Table 6?
5. No ablation experiments on the two proposed losses.


**Summary Of The Paper:**

This paper proposed an unlearnable strategy, TUE. It aims to simultaneously enjoy training-wise transferability and data-wise transferability by enhancing linear separability. Experimental results show that the proposed method shows advantages in both transferability settings.


**Summary Of The Review:**

Overall, this work presents a simple, yet seemingly effective unlearnable strategy, and the paper is well written and easy to understand. But there are some deficiencies in the experiment, so I would give score 6 for now.

---

> ### Author Response · Authors · 2022-11-19
> **Response to Reviewer esp8 (1/4)**
>
> Thanks for your insightful and detailed comments! We hope the following responses can help address your concerns.
>
> > **Q1** In section 3, the authors mentioned that this paper focuses on supervised unlearnable examples, which seems to contradict the training-wise transferability of the proposed method.
>
> **A1** Thanks for your thoughtful comment. We have greatly improved the presentation of our paper and added Figure 1 which shows the practical usage procedure of unlearnable examples (UEs) in the updated paper to help clarify your question on the two definitions. As shown in Figure 1, the practical usage procedure of UEs consists of two stages: *generation* and *evaluation*. During the generation stage, data owners generate the UEs to protect their data by adding the perturbations on the original data. During the evaluation stage, unauthorized third parties try to train a model (in supervised or unsupervised way) on the UEs, while UEs can provide the protection by invalidating the models trained on them. The generation stage provides protection, and the evaluation stage tests how effective the protection is. Supervised unlearnable examples (SUE) are defined based on the generation stage, while training-wise transferability refers to the transferability of SUE to protect against different unauthorized training (supervised training and unsupervised training) in the evaluation stage.
>
> In the generation stage, supervised unlearnable examples and unsupervised unlearnable examples refer to whether we generate the unlearnable examples with the guidance of label information or without the guidance of label information. Specifically, supervised unlearnable examples, like EMN, are generated with the guidance of label information, while unsupervised unlearnable examples, like UCL, are generated without the guidance of label information. In our paper, we focus on supervised unlearnable examples, where the label information is used to generate the perturbations. Our method, TUE, uses labels in its second term $L_{CSD}$.
>
> In the evaluation stage, as our experiments in Section 3.2.1 show, although SUE can protect data from supervised learning, they cannot prevent unsupervised learning. This is verified with a simple adversary who first trains a feature extractor $g$ on the protected data then finetunes to obtain a good classifier $h_g$. Training-wise transferability refers to that the generated SUE can not only protect against the supervised learning, but also be transferred to prevent unauthorized unsupervised learning, which is one property of our proposed TUE.

---

> > ### Author Response · Authors · 2022-11-19
> > **Response to Reviewer esp8 (2/4)**
> >
> > > **Q2** The first term of equation 5 combines unsupervised loss explicitly, while the second term $L_S$ combines supervised loss implicitly. What if we consider both unsupervised and supervised loss explicitly?
> >
> > **A2** Thanks for your good suggestion. *Combining the loss of EMN and UCL explicitly* (EMN-UCL) is a simple and direct method, but we would like to show that EMN-UCL will lose a part of the linear separability of the perturbations, and thus reduce the data-wise transferability and protection against supervised training by the experiments in Table 8 in Appendix D.1. As we analyzed in Section 3.2.2 and Section 5.3, EMN is designed to reduce the supervised loss on target data, while TUE creates the shortcut by linear separability of perturbations, which is not tied to the specific data. As such, EMN-UCL should have *limited* data-wise transferability compared with TUE.
> >
> > In Table 8, the experimental results are given to show the following weaknesses of EMN-UCL:
> >
> > - First, EMN-UCL has limited data-wise transferability, especially when being transferred onto non-target datasets. Because of the poor linear separability from EMN, the accuracies of EMN-UCL on non-target datasets are much higher on almost every dataset and backbone. EMN-UCL with SimCLR backbone even has an accuracy of 79.25% when transferred from CIFAR-10 to SVHN and an accuracy of 45.61% when transferred from CIFAR-10 to CIFAR-100. However, TUE can reduce the accuracy on SVHN to around 12% and the accuracy on CIFAR-100 to less than 11% on every backbone.
> > - Second, EMN-UCL has less protection against supervised training than TUE (see column 3 in Table 8). On CIFAR-10, TUE can reduce the test accuracy of unauthorized supervised training to around 10%, while EMN-UCL has a higher accuracy which is from 13% to 19 % on different backbones. This means EMN-UCL provides less protection against supervised training than TUE.
> >
> > In conclusion, TUE can provide better supervised unlearnability and better data-wise transferability than simply combining unsupervised and supervised loss explicitly.
> >
> > [*Table 8*: Transferability of EMN-UCL, an explicit combination of the supervised and unsupervised losses.]
> >
> > | Unsupervided backbone | Method | Training-wise transferability |  | Data-wise transferability |  |  |  |
> > | --------------------- | -------- | ---------- | ------------ | --------------- | --------------- | --------------- | -------------------- |
> > |  |  | supervised | unsupervised | intraclass swap | interclass swap | CIFAR10 -> SVHN | CIFAR10 -> CIFRA-100 |
> > | SimCLR                | EMN-UCL | 19.89      | 47.83        | 16.15           | 14.82           | 79.25           | 45.61                |
> > |                       | TUE     | 10.67      | 52.38        | 10.16           | 10.9            | 11.72           | 10.53                |
> > | MoCo                  | EMN-UCL | 13.34      | 50.62        | 13.61           | 11.9            | 28.52           | 27.43                |
> > |                       | TUE     | 10.06      | 63.38        | 12.04           | 8.57            | 13.95           | 8.32                 |
> > | SimSiam               | EMN-UCL | 15.36      | 32.74        | 16.15           | 14.82           | 11.14           | 21.47                |
> > |                       | TUE     | 10.03      | 35.57        | 10.25           | 10.47           | 12.93           | 5.10                 |

---

> > > ### Author Response · Authors · 2022-11-19
> > > **Response to Reviewer esp8 (3/4)**
> > >
> > >
> > > > **Q3** It is not clear how much interpolation affects data-wise transferability
> > >
> > > **A3** Thanks for your comment about interpolation. Please first allow us to kindly clarify the typos in the Eq. 7 and Eq. 8 for interpolation. This may cause you to think that the interpolation is conducted between two samples. The interpolation is conducted between two (unlearnable) perturbations, rather than two samples. Specifically, in Eq. 7, $\boldsymbol{x}$ should be corrected to $\boldsymbol{\delta}$, and the correct version of Eq. 7 should be
> > > $$
> > > \boldsymbol{\delta}_k^*=\alpha \boldsymbol{\delta}_i+(1-\alpha) \boldsymbol{\delta}_j \text {, where } y_i \neq y_j
> > > $$
> > > Eq. 8 should also be corrected as
> > > $$
> > > \boldsymbol{\delta}_k^*=\alpha \boldsymbol{\delta}_i+(1-\alpha) \boldsymbol{\delta}_j, \text { where } y_i=y_j
> > > $$
> > > With this correction, we can continue to explain the interpolation clearly. Interpolation of perturbations can help to increase the number of transferable perturbations, when we want to transfer current unlearnable perturbations to a new but larger dataset. The impact of interpolation to data-wise transferability can be checked from two aspects: 1) the change of the unlearnable effect when transferred to the other dataset; and 2) the change of linear separability after interpolation, which can be quantified by CSD. We discuss these two aspects as follows.
> > >
> > > - The first aspect has already been presented in Table 6 in Section 5.3. We can see that, by interpolation, the unlearnable effect is maintained after being transferred from CIFAR10 to CIFAR100 and SVHN.
> > >
> > > [*Table 6*: Test accuracy (%) of transferring the perturbations generated on CIFAR-10 to different datasets.]
> > >
> > > |              | SVHN-small | CIFAR100 | SVHN  |
> > > | ------------ | -------- | -------- | -------- |
> > > | EMN          | 27.59      | 21.80    | 24.72 |
> > > | SN           | 9.58       | 9.35     | 7.77  |
> > > | TUE(SimCLR)  | 9.77       | 10.53    | 11.72 |
> > > | TUE(MoCo)    | 11.29      | 8.32     | 13.95 |
> > > | TUE(SimSiam) | 10.28      | 5.10     | 12.93 |
> > >
> > > - For the second aspect, we show the change of linear separability after the interpolation in Table 9 in Appendix D.2. The interpolation from CIFAR10 to SVHN has almost no change to CSD. Although the interpolation from CIFAR10 to CIFAR100 increases CSD a little, it is still low enough for good linear separability. The change for SN and TUE in CIFAR100 can be almost ignored. (The reason why CSD increases in CIFAR100 is that a larger number of classes would make the perturbation space more crowded.)
> > >
> > > [*Table 9*: The change of the CSD value after interpolation]
> > > |              | CIFAR10 | CIFAR10 -> SVHN | CIFAR10 -> CIFAR100 |
> > > | ------------ | ------- | --------------- | ------------------- |
> > > | EMN          | 1.7404  | 1.6066          | 2.0615              |
> > > | SN           | 0.5584  | 0.6155          | 0.6622              |
> > > | TUE(SimCLR)  | 0.7149  | 0.7298          | 0.8507              |
> > > | TUE(MoCo)    | 0.5276  | 0.5842          | 0.6265              |
> > > | TUE(SimSiam) | 0.6704  | 0.6921          | 0.7970              |
> > >
> > > In conclusion, the interpolation has no impact on the data-wise transferability of TUE. We hope this can clarify your concerns about the impact of interpolation to data-wise transferability.

---

> > > > ### Author Response · Authors · 2022-11-19
> > > > **Response to Reviewer esp8 (4/4)**
> > > >
> > > > > **Q4** What are the backbones of EMN and SN in Table 5 and Table 6?
> > > >
> > > > **A4** Thanks for your question about the details. First, all the methods are generated on the architecture of ResNet-18, except SN which is manually sampled from a distribution. Second, as for the backbones in Table 5 and Table 6, the reason why we label backbone for TUE in the tables is that the loss of TUE includes a Contrastive-Learning (CL) term which is slightly different between different CL backbones. However, EMN and SN do not have such a CL term. Specifically, EMN is generated by inducing perturbations to reduce the cross-entropy loss in a supervised setting. For SN, as we mentioned in Section 1, it is sampled from a manually designed distribution and is not targeting any architecture. SN only focuses on linear separability. It first sets $M$ centers in the perturbation space, where $M$ is the number of protected classes. The generation process of SN can be seen as sampling the perturbations for one class from a Gaussian distribution whose mean is one of the $M$ centers. We have also added the details about baseline methods in Appendix B.1.
> > > >
> > > >
> > > > > **Q5** No ablation experiments on the two proposed losses.
> > > >
> > > > **A5** Thanks for your suggestion. We have run ablation experiments on the two terms of the proposed loss. The results are reported in Tables 10 and 11 in Appendix D.3, which show that removing either term will greatly hurt the performance of TUE.
> > > >
> > > > Particularly, in Table 10, we show that, if we only use CSD loss, the protection against unsupervised training will fail. In Table 11, we show that if only the Contrastive-Learning term is used (reduces to the UCL baseline), it will lose its protection against supervised training.
> > > >
> > > > [*Table 10*: Test accuracy of ResNet-18 trained under different settings on different unlearnable examples (Clean vs. CSD). ]
> > > >
> > > > |          |       | supervised | SimCLR | MoCo  | SimSiam |
> > > > | -------- | ----- | ---------- | ------ | ----- | ------- |
> > > > | CIFAR10  | Clean | 93.79      | 90.04  | 89.9  | 90.59   |
> > > > |          | CSD   | 9.99       | 80.33  | 79.24 | 81.20   |
> > > > | CIFAR100 | Clean | 74.49      | 63.68  | 63.03 | 64.69   |
> > > > |          | CSD   | 1.03       | 55.69  | 51.81 | 58.99   |
> > > >
> > > > [*Table 11*: Test accuracy of ResNet-18 trained under different settings on different unlearnable examples (Clean vs. UCL).  ]
> > > >
> > > > |         |       | CIFAR10    |              | CIFAR100   |              |
> > > > | ------- | ----- | ---------- | ------------ | ---------- | ------------ |
> > > > |         |       | supervised | unsupervised | supervised | unsupervised |
> > > > | SimCLR  | Clean | 93.79      | 90.04        | 74.49      | 63.68        |
> > > > |         | UCL   | 92.86      | 47.78        | 72.17      | 16.68        |
> > > > | MoCo    | Clean | 93.79      | 89.9         | 74.49      | 63.03        |
> > > > |         | UCL   | 92.62      | 44.24        | 71.59      | 18.74        |
> > > > | SimSiam | Clean | 93.79      | 90.59        | 74.49      | 64.69        |
> > > > |         | UCL   | 93.5       | 30.43        | 71.84      | 4.64         |

---

> > > > > ### Comment · Reviewer_esp8 · 2022-11-29
> > > > > **Thank you for the responses**
> > > > >
> > > > > Thank you for the responses and additional experiments to address my concerns. The responses have addressed most of my concerns. A minor comment is how to transfer if the dimensions of the two datasets differ. E.g. how to transfer the perturbation of SVHN to MNIST, I suggest adding a corresponding experiment.

---

> > > > > > ### Author Response · Authors · 2022-12-01
> > > > > > **Response to Reviewer esp8**
> > > > > >
> > > > > > Thanks for your valuable feedback. Please find our response below for your concerns.
> > > > > >
> > > > > > > **Q** Thank you for the responses and additional experiments to address my concerns. The responses have addressed most of my concerns. A minor comment is how to transfer if the dimensions of the two datasets differ. E.g. how to transfer the perturbation of SVHN to MNIST, I suggest adding a corresponding experiment.
> > > > > >
> > > > > > **A** Thanks very much for your timely reply and the insightful suggestion. We have run an additional experiment on transferring perturbations from CIFAR10 (32 $\times$ 32) to STL10 (96 $\times$ 96) and MNIST-M (28 $\times$ 28). STL10 contains 5,000 training samples and 8,000 test samples, while MNIST-M contains 60,000 training samples and 10,000 test samples. The difference between MNIST-M and MNIST is that MNIST-M has figures in colored backgrounds which are in RGB channels. Since the TUE perturbations are generated in RGB channels on CIFAR10 and the data that people release in public (like social media) are usually in RGB channels, we consider transferring it to MNIST-M instead of grayscale MNIST. All experiments are conducted with ResNet18.
> > > > > >
> > > > > > *Experimental details.* When transferring to a dataset with a smaller image size like MNIST-M, we crop part of the perturbations to match the target size. When transferring to a dataset with a larger image size, we repeat the perturbations to cover the whole area of larger images. For example, when transferring from CIFAR10 (32 $\times$ 32) to STL10 (96 $\times$ 96), we repeat the perturbations 3 times in column and 3 times in rows. In this way, one perturbation with 32 $\times$ 32 pixels is repeated 3 $\times$ 3 times to cover a 96 $\times$ 96 size image.
> > > > > >
> > > > > > The results are in Table 16 below, where it shows that our TUE can hold the data-wise transferability across datasets with different image sizes. Particularly, TUE reduces the test accuracy on STL10 from 70.59% to 21.89%~33.72%, and from 99.12% to around 10% on MNIST-M. TUE performs much better than EMN, especially on MNIST-M. EMN can only reduce the test accuracy to 79.67% on MNIST-M. SN demonstrates the best transferability in this case, but it fails the training-wise transferability as we have shown in Section 5.2. These results verify that the unlearnable effect of our TUE can be easily transferred to datasets of different dimensions.
> > > > > >
> > > > > > [*Table 16*. Data-wise transferability on datasets with different image sizes.]
> > > > > >
> > > > > > | Dataset | Clean | EMN   | SN    | TUE (SimCLR) | TUE (MoCo) | TUE (SimSiam) |
> > > > > > | ------- | ----- | ----- | ----- | ------------ | ---------- | ------------- |
> > > > > > | STL10   | 70.59 | 36.39 | 17.57 | 21.89        | 33.1       | 33.72         |
> > > > > > | MNIST-M | 99.12 | 79.67 | 13.43 | 13.85        | 8.58       | 10.03         |

---

> > > > > > > ### Comment · Reviewer_esp8 · 2022-12-02
> > > > > > > **Thank you for the responses**
> > > > > > >
> > > > > > > I appreciate the authors' efforts in the rebuttal, including more experimental results, and clarifications. The responses have addressed my concerns, and I have improved my score accordingly. I encourage the authors to incorporate the additional contents of the rebuttal into the main paper.

---

> > > > > > > > ### Author Response · Authors · 2022-12-02
> > > > > > > > **Response to Reviewer esp8**
> > > > > > > >
> > > > > > > > We are very fortunate to improve the paper with your help and will put the additional contents of the rebuttal into the main paper. Again, thank you very much for reviewing our paper and giving suggestions.

---

### Official Review · Reviewer_fZkw · 2022-11-07

**Confidence:** 3
**Correctness:** 3
**Technical Novelty And Significance:** 3
**Empirical Novelty And Significance:** 3
**Recommendation:** 6

**Clarity, Quality, Novelty And Reproducibility:**

- The proposed method is described relatively clearly, but the organization of the paper could be improved.
    - The text is occasionally redundant, and some points are mentioned multiple times before sufficiently defined or described. Adding forward references in the introduction or related work to sections later may help improve the overall organization.
    - It can be difficult to find some model and training details and would be difficult to reproduce some experiments with the details provided (Section 3.2.1 or the compared methods, for example).
- The motivation is not completely clear to me; the experiments seem to aim preventing learnability in general, but the stated application is preventing training on protected user data, which would rarely be the full set of available training data. It is not clear to me that the proposed method would be effective in such a case where the data is only partly obscured. The unsupervised case is also stated as left to future work on page 2 but is given significant focus.
- It is not clear to me how the interpolations for new classes are validated in Section 5.3 beyond overall transfer performance.
- Why does UCL have no protection in supervised settings? While the other two baselines are discussed in Section 3, this is not; it would be nice to add a short description.
- Minor presentation notes: the space has been overly compressed around tables and figures (particularly captions); the labels in Figure 1 are too small to read; there are occasional typos (Unlearnbale, unsuperivsed, unlearnbility); \citet and \citep are often flipped.

**Strength And Weaknesses:**

Strengths:
- Experiments on three datasets support that the proposed method generally improves both unlearnability and improves unlearnability transfer compared to recent methods (by a large margin in some cases, but certainly more consistently across different settings).

Weaknesses:
- The motivation is somewhat unclear at points. Paper clarity and organization could be improved.


**Summary Of The Paper:**

This paper proposes a strategy to address a shortcoming of prior 'unlearnable' strategies which aim to prevent training on unauthorized, namely the lack of transferability across training settings and datasets. Typical strategies add perturbations generated for a specific training setting and target dataset, and so typically do not transfer. This paper proposes a strategy called Transferable Unlearnable Examples (TUE) based on Classwise Separability Discriminant (CSD) to enhance linear separability of perturbation classes to improve the transferability of unlearnable examples. This method is an extension of prior contrastive learning algorithms to also balance linear separability, with interpolation for coverage of missing classes. Experiments show its improvements over other recent methods in most cases, particularly transferability.

**Summary Of The Review:**

This paper presents an effective strategy to promoting transferable unlearnability in image classification settings. While the performance improvements from the method are encouraging, particularly for transfer, the writing of the paper itself could be improved.

---

> ### Author Response · Authors · 2022-11-19
> **Response to Reviewer fZkw (1/4)**
>
> Thanks for your valuable comments. Please kindly find our response below for your questions.
>
> > **Q1** The motivation is not completely clear to me; the experiments seem to aim at preventing learnability in general, but the stated application is preventing training on protected user data, which would rarely be the full set of available training data. It is not clear to me that the proposed method would be effective in such a case where the data is only partly obscured.
>
> **A1** Thanks for your question. In some cases, we have access to the whole dataset. Taking social media platforms as an example, for the company, they can protect all its users' data that is released on the social media and they do have access to all the data. For users, they can protect their own data by perturbing before uploading. In our paper, we focus on the cases where the protection can be added onto all the data.
>
> In terms of the suggested partial protection, we also evaluate when people can only perturb part of the classes of the data (like the users can only perturb their own data) by the experiments shown in Table 13 in Appendix D.5. From Table 13, we can see that although the unsupervised training can still learn semantic features from the partly perturbed data, the data is protected from supervised training very effectively. The accuracy on perturbed classes is almost 0 for most cases. When we only have access to a part of the data, the unlearnable examples can still protect that part of classes.
>
> [*Table 13*: Test accuracy (%) on partly perturb datasets. The accuracy is calculated on the perturbed classes.]
>
> |                           | CIFAR10    |              |            |              | CIFAR100   |              |            |              |
> | ------------------------- | ---------- | ------------ | ---------- | ------------ | ---------- | ------------ | ---------- | ------------ |
> | number of perturbed class | 1          |              | 2          |              | 1          |              | 10         |              |
> |                           | supervised | unsupervised | supervised | unsupervised | supervised | unsupervised | supervised | unsupervised |
> | TUE(SimCLR)               | 0.20       | 93.4         | 0.15       | 92.58        | 4.00       | 88.00        | 0.90       | 69.00        |
> | TUE(MoCo)                 | 0.00       | 94.4         | 0.05       | 95.55        | 4.00       | 83.30        | 0.40       | 73.30        |
> | TUE(SimSiam)              | 0.10       | 82.7         | 0.05       | 86.15        | 7.00       | 83.00        | 0.20       | 55.60        |

---

> > ### Author Response · Authors · 2022-11-19
> > **Response to Reviewer fZkw (2/4)**
> >
> > > **Q2** The unsupervised case is also stated as left to future work on page 2 but is given significant focus.
> >
> > **A2** Please allow us to clarify the concepts of unsupervised unlearnable examples and unauthorized unsupervised learning that make the reviewer confused. We have added Figure 1, which shows the practical usage procedure of unlearnable examples (UEs), in the updated paper to help clarify your question on the two definitions. As shown in Figure 1, the practical usage procedure of UEs consists of two stages: *generation* and *evaluation*. During the generation stage, data owners generate the UEs to protect their data by adding the perturbations on the original data. During the evaluation stage, unauthorized third parties try to train a model (in supervised or unsupervised way) on the UEs, while UEs can provide the protection by invalidating the models trained on them. The generation stage provides protection, while the evaluation stage tests how effective the protection is. Supervised unlearnable examples (SUE) are defined based on the generation stage, while unauthorized unsupervised learning refers to one of the unauthorized trainings in the evaluation stage.
> >
> > In the generation stage, supervised unlearnable examples and unsupervised unlearnable examples refer to whether we generate the unlearnable examples guided by label information or not. Specifically, supervised unlearnable examples, like EMN, are generated with the guidance of label information, while unsupervised unlearnable examples, like UCL, are generated without the guidance of label information. In our paper, we focus on supervised unlearnable examples, where the label information is used to generate the perturbations and leave the unsupervised unlearnable examples to the future work. Our method, TUE, uses labels in its second term ($L_{CSD}$) and is one of the supervised unlearnable examples.
> >
> > In the evaluation stage, the unauthorized third parties may use supervised or unsupervised training to learn from the data. As our experiments in Section 3.2.1 show, although SUE can protect data from supervised learning, they cannot prevent unsupervised learning. This is verified with a simple adversary who first trains a feature extractor $g$ on the protected data then finetunes to obtain a good classifier $h_g$. Training-wise transferability refers to that the generated SUE can not only protect against the supervised learning, but also be transferred to prevent unauthorized unsupervised learning, which is one property of our proposed TUE.

---

> > > ### Author Response · Authors · 2022-11-19
> > > **Response to Reviewer fZkw (3/4)**
> > >
> > > > **Q3** It is not clear to me how the interpolations for new classes are validated in Section 5.3 beyond overall transfer performance.
> > >
> > > **A3** Thanks for your comment about interpolation. Please first allow us to kindly clarify the typos in the Eq. 7 and Eq. 8 for interpolation. This may cause you to think that the interpolation is conducted between two samples. The interpolation is conducted between two (unlearnable) perturbations, rather than two samples. Specifically, in Eq. 7, $\boldsymbol{x}$ should be corrected to $\boldsymbol{\delta}$​, and the correct version of Eq. 7 should be
> > > $$
> > > \boldsymbol{\delta}_k^*=\alpha \boldsymbol{\delta}_i+(1-\alpha) \boldsymbol{\delta}_j \text {, where } y_i \neq y_j
> > > $$
> > > Eq. 8 should also be corrected as
> > > $$
> > > \boldsymbol{\delta}_k^*=\alpha \boldsymbol{\delta}_i+(1-\alpha) \boldsymbol{\delta}_j, \text { where } y_i=y_j
> > > $$
> > > With this correction, we can continue to explain the interpolation clearly.
> > >
> > > As shown in Sections 4 and 5.3, we used interpolation to create more classes of perturbations to transfer TUE from CIFAR10 to CIFAR100, i.e., 90 new classes of perturbations are created to protect CIFAR100, while the remaining 10 classes are directly transferred from CIFAR10. To better understand how the unlearnable effect on interpolated new classes is, we show it from two perspectives, i.e., the unlearnable effect against unauthorized training and linear separability (quantified by CSD). First, in Table 14 in Appendix D.6, we show that the unlearnable effect on the new created classes is as good as overall classes. Second, in Table 15 in Appendix D.6, we show that the linear separability of interpolated perturbations for new classes is almost the same as the linear separability for overall classes. In conclusion, the interpolated new classes have good linear separability and the unlearnable effect can transfer well on non-target dataset.
> > >
> > > [*Table 14*: Test accuracy (%) on new classes from interpolation]
> > >
> > > | CIFAR10 -> CIFAR100 | new classes | overall classes |
> > > | ------------------- | ----------- | --------------- |
> > > | TUE(SimCLR)         | 6.41        | 10.53           |
> > > | TUE(MoCo)           | 7.35        | 8.32            |
> > > | TUE(SimSiam)        | 11.94       | 5.10            |
> > >
> > > [*Table 15*: Linear separability on new classes from interpolation]
> > >
> > > | CIFAR10 -> CIFAR100 | CSD of new classes | CSD of overall classes |
> > > | ------------------- | ------------------ | ---------------------- |
> > > | TUE(SimCLR)         | 0.8527             | 0.8507                 |
> > > | TUE(MoCo)           | 0.6281             | 0.6265                 |
> > > | TUE(SimSiam)        | 0.7979             | 0.797                  |
> > >
> > >
> > >
> > >
> > > > **Q4** Why does UCL have no protection in supervised settings? While the other two baselines are discussed in Section 3, this is not; it would be nice to add a short description.
> > >
> > > **A4** Thanks for your question. We have mentioned this in the discussion of Figure 6 in Section 5.4. The main reason why UCL has no protection in supervised settings is that the perturbations generated by UCL do not have linear separability as shown in Figure 6, which is the easy-to-learn feature to invalidate supervised training. This is caused by the missing of label information during the generation process. So for a supervised model, UCL is not a helpful feature and will not vanquish the semantic feature.

---

> > > > ### Author Response · Authors · 2022-11-19
> > > > **Response to Reviewer fZkw (4/4)**
> > > >
> > > > > **Q5** The proposed method is described relatively clearly, but the organization of the paper could be improved. 1) The text is occasionally redundant, and some points are mentioned multiple times before sufficiently defined or described. Adding forward references in the introduction or related work to sections later may help improve the overall organization. 2) It can be difficult to find some model and training details and would be difficult to reproduce some experiments with the details provided (Section 3.2.1 or the compared methods, for example).
> > > >
> > > > **A5** Thanks for your suggestion. We have greatly improved our representation and organization in the updated submission as you suggested.
> > > >
> > > > In terms of the second point, the details to reproduce the experiment can be found at Section 5.1 and Appendix B. We will also release the code in the future. The settings of experiment in Section 3.2.1 is the same as the evaluation settings in Section 5.1. Following your suggestion, we also briefly mentioned the settings of experiment in Section 3.2.1 for better understanding, and provide more details about the baseline methods in Appendix B.1. We also attach the baseline details below for convenience.
> > > >
> > > > *EMN* generates the easy-to-learn feature by inducing the perturbations to reduce the supervised cross-entropy loss via alternating optimization on the bi-level min-min problem:
> > > > $$
> > > > \min_\theta \min_{\boldsymbol{\delta_i} \in \\{ \boldsymbol{v}: {|\boldsymbol{v}|}_{\infty} \leq \epsilon\\}} \sum_1^n \mathcal{L} ( f_\theta (\boldsymbol{x}_i + \boldsymbol{\delta}_i), y_i )
> > > > $$
> > > > The outer minimization can imitate the training process, while the inner minimization can induce $\boldsymbol{\delta}_i$ to have the property of minimizing the supervised loss. Due to this property, deep models will pay more attention to the easy-to-learn $\boldsymbol{\delta}_i$ and ignore $\boldsymbol{x}_i$.
> > > >
> > > > *UCL* also generates perturbations which is easy-to-learn for unsupervised contrastive learning by inducing the perturbations to reduce the contrastive loss via alternating optimization on this bi-level min-min problem:
> > > > $$
> > > > \min_\theta \min_{\boldsymbol{\delta_i} \in \\{ \boldsymbol{v}: {|\boldsymbol{v}|}_{\infty} \leq \epsilon\\}} \sum_1^n \mathcal{L}_\text{CL} \Big ( f \big(\theta, T_1(\boldsymbol{x}_i + \boldsymbol{\delta}_i)\big), f \big(\theta, T_2(\boldsymbol{x}_i + \boldsymbol{\delta}_i)\big) \Big )
> > > > $$
> > > > It does not require label information and can work well on different contrastive learning backbones.
> > > >
> > > > *SN* is sampled from a manually designed distribution. It is not targeting any backbone or dataset and only focuses on linear separability. SN first sets $M$ centers in the perturbation space, which are far away from each other. $M$ is the number of protected classes. The generation process of SN can be seen as sampling the perturbations for one class from a Gaussian distribution whose mean is one of the $M$ centers.
> > > >
> > > >
> > > >
> > > > > **Q6** Minor presentation notes: the space has been overly compressed around tables and figures (particularly captions); the labels in Figure 1 are too small to read; there are occasional typos (Unlearnbale, unsuperivsed, unlearnbility); \citet and \citep are often flipped.
> > > >
> > > > **A6** Thanks for pointing this out. We have greatly improved our representation and fixed the spacing, formatting and grammar mistakes.

---

> ### Comment · Reviewer_fZkw · 2022-12-07
> **Response to authors**
>
> Thank you for your detailed response!
> I acknowledge that I have read the response and that it did address some of my concerns and points of confusion.
> I would like to clarify that I had not responded after the authors’ response the 18th because the open discussion period ended Nov 18th.
>
> Two (small and addressable) points for the final version:
>
> **Interpolation to new classes**: It seems to me that the interpolated perturbations are separable by construction? so the separability of the interpolated perturbations alone would not validate that interpolation has the effect it is intended to have on new classes. Please clarify if I have misunderstood.
> It would be helpful to present a clean baseline on the new data in addition to the various unlearnable effects methods in 5.3 and D.6 to validate the impact of the methods; without this, though we assume these methods perform well without perturbations, it is not clear the potential impact and differences between the different methods (maybe the new data is not transferable separable without perturbations). It would be nice for context to see this baseline.
>
> **Organization**: The additional materials in the response and appendices are helpful and largely clarifying. I would encourage the authors to add some of this content to the main text for the final version.
> Though the authors said the organization and formatting have been greatly improved, the body of the paper was minimally changed, and some of the new appendix sections are not referenced in the main text. This is very understandable for the response period but I encourage the authors to bring some of the additional content to the body of the paper and to edit for organization and formatting (order of presentation, space compression, appendix references, typos). For example on page 2 it says unsupervised unlearnable examples are left to future though unsupervised examples are part of the following discussion and results, and some concepts are referenced times before they are defined (ex data-wise transferability). Typos and space compression do not appear to have changed (resultign, evalaution, unlearnbility, ..)
> These are not so crucial but I believe they will improve the final version.
>
> I maintained my score but am not opposed to the paper being accepted if the other reviewers and chairs are in favor of its acceptance.
> I note that my confidence is lower than the other reviewers'.

---

> > ### Author Response · Authors · 2022-12-08
> > **Response to Reviewer fZkw (1/2)**
> >
> > Thanks for your positive feedback to our rebuttal! We are glad that we have addressed most of your concerns. We checked the key dates of ICLR23 and it seems that Nov 18 is the deadline for paper revision and open discussion is allowed until Dec 12. Therefore, we still have time for potential discussions. Feel free to let us know if you have further concerns. Below are our responses to your two (small and addressable) points.
> >
> > > **Q1** It seems to me that the interpolated perturbations are separable by construction? so the separability of the interpolated perturbations alone would not validate that interpolation has the effect it is intended to have on new classes. Please clarify if I have misunderstood.
> >
> > **A1** The interpolated perturbations are constructed to be linear separable because linear separability is the key reason for data-wise transferability and unlearnability against supervised training. This is verified in Section 5.3, Appendix D.2, D.6 and [1]. Particularly, the linear separable perturbations can distract the model from learning semantic features. It means that the linear separability plays an important role in maintaining the unlearnable effect on new classes. In Figure 5 and Appendix D.6, it is also shown that the interpolated perturbations for new classes can hold such linear separability well. In conclusion, good linear separability from interpolation maintains the data-wise transferable unlearnability on new classes.
> >
> > > **Q2** It would be helpful to present a clean baseline on the new data in addition to the various unlearnable effects methods in 5.3 and D.6 to validate the impact of the methods; without this, though we assume these methods perform well without perturbations, it is not clear the potential impact and differences between the different methods (maybe the new data is not transferable separable without perturbations). It would be nice for context to see this baseline.
> >
> > **A2** Thanks for your advice. It is indeed helpful if we add the clean baseline especially in Append D.6, which can make the conclusion on new classes clearer. The clean baseline for Section 5.3 and Append D.6 is shown in Table 17. We will add the clean baseline to Table 5 and Table 6 in Section 5.3 and Table 14 in Appendix D.6.
> >
> > [*Table 17*: Clean baselines]
> >
> > | | CIFAR-10 | CIFAR-100 | SVHN | SVHN-small | new classes of CIFAR-100 in Table 14|
> > | - | - | - | - | - | - |
> > | clean | 93.79 |74.49| 96.67|96.54| 74.21|
> >
> > > **Q3** For example on page 2 it says unsupervised unlearnable examples are left to future though unsupervised examples are part of the following discussion and results.
> >
> > **A3** The two concepts of Unsupervised Unlearnable Examples and unauthorized unsupervised training may be confusing. We leave the Unsupervised Unlearnable Examples to the future and discuss unauthorized unsupervised training in this paper. We added Figure 1 to help clarify the two concepts. Figure 1 shows the practical usage procedure of unlearnable examples (UEs) which consists of two stages: *generation* and *evaluation*. During the generation stage, data owners generate the UEs to protect their data by adding the perturbations on the original data. During the evaluation stage, unauthorized third parties try to train a model (in supervised or unsupervised way) on the UEs, while UEs can provide the protection by invalidating the models trained on them. The generation stage provides protection, while the evaluation stage tests how effective the protection is. Supervised unlearnable examples (SUE) are defined based on the generation stage, while training-wise transferability refers to the transferability of SUEs to protect against different unauthorized training (supervised training and unsupervised training) in the evaluation stage.
> >
> > In the generation stage, supervised unlearnable examples and unsupervised unlearnable examples refer to whether we generate the unlearnable examples guided by label information or not. Specifically, supervised unlearnable examples, like EMN, are generated with the guidance of label information, while unsupervised unlearnable examples, like UCL, are generated without the guidance of label information. In our paper, we focus on supervised unlearnable examples, where the label information is used to generate the perturbations. Our method, TUE, uses labels in its second term ($L_{CSD}$).
> >
> > In the evaluation stage, as our experiments in Section 3.2.1 show, although SUEs can protect data from supervised learning, they cannot prevent unsupervised learning. This is verified with a simple adversary who first trains a feature extractor $g$ on the protected data then finetunes to obtain a good classifier $h_g$. Training-wise transferability refers to that the generated SUE can not only protect against the supervised learning, but also be transferred to prevent unauthorized unsupervised learning, which is one property of our proposed TUE.

---

> > > ### Author Response · Authors · 2022-12-08
> > > **Response to Reviewer fZkw (2/2)**
> > >
> > > > **Q4** Paper organization.
> > >
> > > **A4** Thanks for your detailed suggestions on paper organization (especially mentioning the new contents of Appendix in the main text). We will follow your suggestions to improve the organization for the final version.
> > >
> > > Thanks again for your positive feedback to our rebuttal. We hope that the above can clarify your two minor concerns. If so, we would greatly appreciate it if you could consider raising the score. We are very fortunate to further improve the paper with your help. As aforementioned, the discussion can continue until Dec 12. Therefore, feel free to let us know if you have further concerns.
> > >
> > > [1] Yu, Da, et al. "Availability attacks create shortcuts." Proceedings of the 28th ACM SIGKDD Conference on Knowledge Discovery and Data Mining. 2022.

---

> > > > ### Comment · Reviewer_fZkw · 2022-12-08
> > > > **Response to authors**
> > > >
> > > > Thank you for the quick response. It would be great to see the numbers from the table you've added here added to the paper.
> > > > With the consistent improvements I will increase my score slightly.
> > > > I encourage you to add the content from these responses to the final paper.
> > > >
> > > > A note for future that the open discussion period between public/authors/reviewers Nov 4-18 is separate from the reviewer/AC/PC discussion period Nov 18-Dec 12, though the portal remains open during that time if they have questions for the authors. Repeated pings to the reviewers during the latter period from the authors may not be received as intended.

---

> > > > > ### Author Response · Authors · 2022-12-08
> > > > > **Thank you for the responses**
> > > > >
> > > > > Thanks for your comments and suggestions! We will continue to improve our paper in the final version and will put the additional contents of the rebuttal into the main paper. Again, thank you very much for reviewing our paper and giving suggestions.

---

### Author Response · Authors · 2022-11-19
**Summary of the major revision**

We thank the reviewers for the thorough and detailed reviews on our submission. We summarize major changes that we have made below. All changes are marked in blue in the updated submission.
- We provided Figure 1 in Section 1 to clarify the practical usage procedure of unlearnable examples.
- We corrected typos in Eq. 7 and Eq. 8 about interpolation.
- We added more details about experiment settings like backbones in Section 5.1 and baselines in Appendix B.1.
- We provided the visualization of interpolation in Appendix C.
- We compared our TUE with simply combining EMN and UCL in Appendix D.1 and Supervised Contrastive Learning in Appendix D.4.
- We provided the experiment on the change of CSD after interpolation in Appendix D.2 and the performance of new classes from interpolation in Append D.6.
- We provided ablation studies on the two terms of the proposed loss in D.3.
- We did experiment on the cases where only part of the dataset can be perturbed in Appendix D.5.

---

### Decision · Program_Chairs · 2023-01-20

**Decision:**

Accept: poster

**Justification For Why Not Higher Score:**

Results are convincing, especially after the extensive added experiments, but presentation/organization could be better.

**Justification For Why Not Lower Score:**

Results are convincing and experiments are quite thorough, especially after the rebuttal.

**Metareview: Summary, Strengths And Weaknesses:**

The study considers the setting of "unlearnable examples", the introduction of appropriately crafted samples that, when combined with real data, distort learning from the (perturbed) dataset. The authors study how to transfer such examples to other datasets, and propose a methodology for doing so.

Strengths of the paper include an interesting setting, a clearly motivated method through Clustering Separability Discriminant analysis, and a very thorough/extensive experimentation method. Reviewers raised several concerns, too many to list here; nevertheless the authors eventually addressed them via the rebuttal process, adding a significant amount of new experiments to improve their paper. Overall, they should be encouraged to add these to the supplement in the final submission. On the negative side, the paper could benefit from polishing, which was found by several reviewers to need improvement.

**Note From Pc:**

if the above contains the word "oral" or "spotlight" please see: "oral" presentation means -> notable-top-5% and "spotlight" means -> notable-top-25%. As stated in our emails, we are disassociating presentation type from AC recommendations

**Summary Of Ac-Reviewer Meeting:**

Unfortunately, only one reviewer attended the meeting, but we had a good discussion on the pros and cons of the paper. In addition, with the exception of one reviewer, who also wrote a terse but positive review, everyone else was engaged in conversation with the authors, so I am fairly confident that their concerns were addressed.